# Redox-controlled reorganization and flavin strain within the ribonucleotide reductase R2b–NrdI complex monitored by serial femtosecond crystallography

Juliane John[1], Oskar Aurelius[1,2], Vivek Srinivas[1], Patricia Saura[1], In-Sik Kim[3], Asmit Bhowmick[3], Philipp S Simon[3], Medhanjali Dasgupta[3], Cindy Pham[3], Sheraz Gul[3], Kyle D Sutherlin[3], Pierre Aller[4,5], Agata Butryn[4,5†], Allen M Orville[4,5], Mun Hon Cheah[6], Shigeki Owada[7,8], Kensuke Tono[7,8], Franklin D Fuller[9], Alexander Batyuk[9], Aaron S Brewster[3], Nicholas K Sauter[3], Vittal K Yachandra[3], Junko Yano[3], Ville RI Kaila[1], Jan Kern[3]*, Hugo Lebrette[1*‡], Martin Högbom[1]*

[1]Department of Biochemistry and Biophysics, Arrhenius Laboratories for Natural Sciences, Stockholm University, Stockholm, Sweden; [2]MAX IV Laboratory, Lund University, Lund, Sweden; [3]Molecular Biophysics and Integrated Bioimaging Division, Lawrence Berkeley National Laboratory, Berkeley, United States; [4]Diamond Light Source Ltd, Harwell Science and Innovation Campus, Didcot, United Kingdom; [5]Research Complex at Harwell, Harwell Science and Innovation Campus, Didcot, United Kingdom; [6]Department of Chemistry - Ångström, Molecular Biomimetics, Uppsala University, Uppsala, Sweden; [7]Japan Synchrotron Radiation Research Institute, Sayo-gun, Japan; [8]RIKEN SPring-8 Center, Sayo-gun, Japan; [9]LCLS, SLAC National Accelerator Laboratory, Menlo Park, United States

*For correspondence:
jfkern@lbl.gov (JK);
hugo.lebrette@dbb.su.se (HL);
hogbom@dbb.su.se (MH)

Present address:
†Macromolecular Machines Laboratory, The Francis Crick Institute, London, United Kingdom; ‡Laboratoire de Microbiologie et Génétique Moléculaires (LMGM), Centre de Biologie Intégrative (CBI), CNRS, UPS, Université de Toulouse, Toulouse, France

Competing interest: The authors declare that no competing interests exist.

**Abstract** Redox reactions are central to biochemistry and are both controlled by and induce protein structural changes. Here, we describe structural rearrangements and crosstalk within the *Bacillus cereus* ribonucleotide reductase R2b–NrdI complex, a di-metal carboxylate-flavoprotein system, as part of the mechanism generating the essential catalytic free radical of the enzyme. Femtosecond crystallography at an X-ray free electron laser was utilized to obtain structures at room temperature in defined redox states without suffering photoreduction. Together with density functional theory calculations, we show that the flavin is under steric strain in the R2b–NrdI protein complex, likely tuning its redox properties to promote superoxide generation. Moreover, a binding site in close vicinity to the expected flavin $O_2$ interaction site is observed to be controlled by the redox state of the flavin and linked to the channel proposed to funnel the produced superoxide species from NrdI to the di-manganese site in protein R2b. These specific features are coupled to further structural changes around the R2b–NrdI interaction surface. The mechanistic implications for the control of reactive oxygen species and radical generation in protein R2b are discussed.

## Editor's evaluation

This paper reports a fundamental set of new results that are obtained using compelling methods in protein crystallography and related fields to investigate and visualize the complex mechanism of an enzyme. The paper will be of interest to a broad audience in structural biology, biochemistry, and enzymology, providing a detailed mechanism of an important biological system and demonstrating

useful tools. The work is timely and has implications for future investigations of complex biochemical processes.

## Introduction

Ribonucleotide reductases (RNRs) are essential metalloenzymes that employ sophisticated radical chemistry to reduce the 2′-OH group of ribonucleotides and thereby produce deoxyribonucleotides (dNTPs), the building blocks of DNA. This reaction is the only known pathway for the *de novo* synthesis of dNTPs. Three classes of RNRs are differentiated based on their structural features and radical generating mechanism. Class I consists of a radical generating subunit, R2 and a catalytic subunit, R1. Oxygen is required to generate a radical in the R2 subunit, which is reversibly shuttled to R1 to initiate the ribonucleotide reduction. Class I, found in eubacteria and all eukaryotes, is to date divided into five subclasses, Ia–Ie, based on the type of metal cofactor, metal ligands, and radical storage in R2 (*Högbom et al., 2020*). The R2 subunit is characterized by a ferritin-like fold housing two metal ions coordinated by six conserved residues (*Nordlund and Reichard, 2006*), that is two histidines and four carboxylates (with the exception of subclass Ie, *Blaesi et al., 2018*; *Srinivas et al., 2018*). The class Ib R2 subunit (R2b) metal site can bind two manganese ions or two iron ions. The two metal ions are oxidized from the M(II)/M(II) state to a short-lived M(III)/M(IV) intermediate which decays to M(III)/M(III) while producing a radical species on an adjacent tyrosyl residue (Tyr·), where it is also stored (*Cotruvo et al., 2013*; *Cotruvo and Stubbe, 2010*). The di-manganese R2b was shown to be the physiologically relevant form of R2b (*Cotruvo and Stubbe, 2010*; *Cox et al., 2010*), it accumulates higher amounts of radical and has higher enzymatic activity than the di-iron R2b. Molecular oxygen ($O_2$) can directly activate the di-iron R2b (*Huque et al., 2000*). In contrast, the di-manganese R2b form cannot react with $O_2$ but requires a superoxide radical ($O_2^{\bullet-}$) provided by NrdI, a flavoprotein which is generally encoded in the same operon as R2b (*Figure 1*; *Berggren et al., 2014*; *Cotruvo and Stubbe, 2010*; *Roca et al., 2008*).

NrdI is a small, globular protein that binds the redox-active flavin mononucleotide (FMN) cofactor. NrdI stands out in comparison to other flavodoxins, which perform single one- or two-electron transfers, by being able to perform two successive one-electron reductions (*Cotruvo and Stubbe, 2008*). Fully reduced NrdI with hydroquinone FMN (NrdI$_{hq}$) can reduce $O_2$ to $O_2^{\bullet-}$ while being oxidized to semiquinone NrdI (NrdI$_{sq}$) (*Cotruvo et al., 2013*). NrdI$_{sq}$ can in turn produce a second $O_2^{\bullet-}$ from another $O_2$ molecule and become fully oxidized (NrdI$_{ox}$) in the process (*Berggren et al., 2014*). The oxidation state of NrdI can be followed by ultraviolet–visible light absorption (UV–vis) spectroscopy. NrdI is faint yellow in the hydroquinone state, dark blue in the semiquinone state, and bright orange in the oxidized state (*Røhr et al., 2010*; *Figure 1*, *Figure 2—figure supplement 1*). NrdI binds tightly to R2b in a 1:1 ratio with FMN at the protein–protein interface, forming a dimer of heterodimers. Previously obtained crystal structures of the R2b–NrdI complex from *Escherichia coli* and *Bacillus cereus* show that a conserved channel connects FMN and the metal site in R2b (*Boal et al., 2010*;

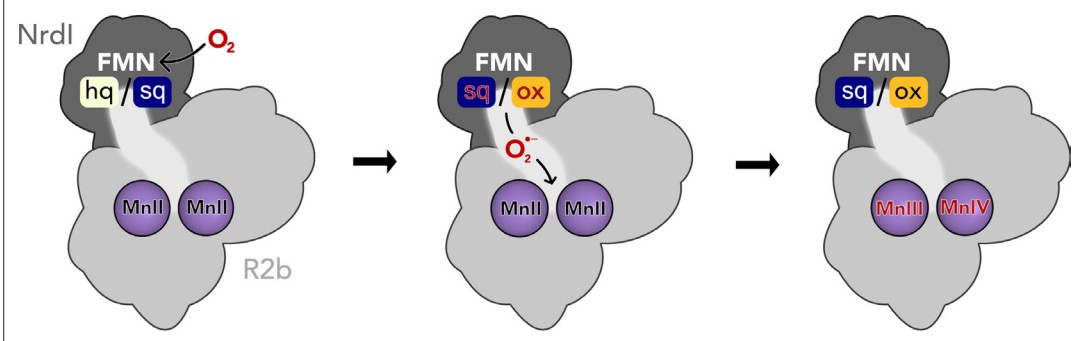

**Figure 1.** Activation of the di-manganese centre in ribonucleotide reductase class Ib R2. Both hydroquinone (hq) and semiquinone (sq) flavin mononucleotide (FMN) of NrdI can reduce molecular oxygen to superoxide, which is shuttled to the metal site in R2b and activates the di-manganese cofactor. FMN is oxidized either to semiquinone or the fully oxidized form (ox) in the process.

*Hammerstad et al., 2014*). This channel is proposed to shuttle $O_2^{•-}$ produced by NrdI in complex with R2b to the metal site, where the radical is formed. Here, we investigate the structural details and reaction mechanism of the activation of NrdI by $O_2$ as well as the subsequent shuttling of $O_2^{•-}$ to the metal site for the generation of Tyr· in R2b.

Studying structural changes in redox-active proteins is challenging from an experimental point of view. X-ray crystallography is a well-suited method for investigating both high-resolution structural details in the redox-active centres and overall reorganization of proteins. However, both metalloenzymes and flavodoxins are very sensitive to photoreduction during the exposure to the high-energy X-ray radiation of synchrotron sources. Synchrotron crystal structures of RNR and NrdI proteins invariably suffer from photoreduction, which complicates the determination of the oxidation state of redox-active centres and has proven to be a problem for obtaining fully oxidized structures (*Grave et al., 2019*; *Johansson et al., 2010*; *Røhr et al., 2010*). While there are well-established experimental workarounds like short exposure times, helical data collection, and serial synchrotron crystallography the time-scales necessary to obtain the diffraction data using these methods cannot fully avoid the reducing effects of the X-ray radiation (*Spence, 2017*). Serial femtosecond crystallography (SFX) overcomes this problem by the so-called diffraction-before-destruction principle (*Doerr, 2011*; *Nass, 2019*; *Neutze et al., 2000*). Short intense X-ray pulses of the duration of only a few femtoseconds are produced by an X-ray free electron laser (XFEL), illuminating one crystal and giving rise to one diffraction pattern at a time. The crystal is generally destroyed in the process and needs to be replaced by a new one for the next image. This method allows to record the scattering information before atomic displacement has time to occur resulting in a dataset effectively free from the effects of radiation damage (*Spence, 2017*).

Here, we present the first SFX structures of R2b in complex with NrdI in the oxidized and hydroquinone state. The datasets reveal redox-dependent structural rearrangements both in the FMN-binding pocket of NrdI and at the protein interface. Despite the significant rearrangement in the direct vicinity of the cofactor, FMN itself changes surprisingly little between redox states. This marks an interesting contrast to structures of free NrdI, which display significant conformational change of FMN between different oxidation states. We conclude that the R2b–NrdI complex formation is restricting FMN movement and inhibits this conformational change with possible implications for its redox properties and $O_2$ reactivity. Moreover, we performed quantum chemical density functional theory (DFT) calculations to computationally estimate the strain energy and additional effects on the FMN redox properties. We also describe the first R2b–NrdI complex with a di-manganese metal centre from *B. cereus*. The metal coordination in both structures blocks access to the channel that connects FMN and the metal site providing further information on gating and control of catalytic reactive oxygen species.

## Results

### SFX crystal structures of *B. cereus* R2b in complex with oxidized and hydroquinone NrdI collected at ambient temperature

An initial crystallization condition for the R2b–NrdI complex was optimized to yield a sufficient amount of crystals smaller than 100 µm in the longest axis (*Figure 2—figure supplement 1*), the maximum acceptable crystal size for the experimental setup. Crystals were initially tested at SACLA (SPring-8 Angstrom Compact free electron LAser, Japan) under aerobic conditions with a grease extruder setup (data not shown) (*Sugahara et al., 2015*; *Tono et al., 2013*). The crystals diffracted to 2 Å and proved to be stable under room temperature for several days and sturdy enough to handle the physical stress of being manipulated for the experiment. At LCLS (Linac Coherent Light Source at SLAC National Accelerator Laboratory, USA), we obtained two structures of the *B. cereus* R2b–NrdI (*Bc*R2b–NrdI) complex in different defined redox states by SFX. The datasets were collected at room temperature under anaerobic conditions using the drop-on-demand sample delivery method (*Fuller et al., 2017*) (see Materials and methods for details).

*Bc*R2b protein was produced metal free to allow full control over metal loading during complex reconstitution. The metal content of the protein was determined by total reflection X-ray fluorescence (TXRF) and only trace amounts of metals could be detected. The iron and manganese content per R2b monomer corresponded to metal-to-protein molar ratios of 0.27% ± 0.04% and 0.07% ± 0.04%, respectively. The *Bc*R2b–NrdI complex was reconstituted *in vitro* by mixing both proteins in a molar

**Table 1.** Data collection and refinement statistics.

Values in parenthesis are for the highest resolution shell.

| | $Bc$R2b$_{MnMn}$–NrdI$_{ox}$ | $Bc$R2b$_{MnMn}$–NrdI$_{hq}$ |
|---|---|---|
| PDB ID | 7Z3D | 7Z3E |
| **Data collection statistics** | | |
| XFEL source | LCLS MFX | LCLS MFX |
| Wavelength (Å) | 1.30 | 1.30 |
| Space group | $C222_1$ | $C222_1$ |
| Unit cell dimensions $a$, $b$, $c$ (Å) | 61.4, 125.6, 145.0 | 61.2, 125.8, 144.8 |
| Unit cell angles $α$, $β$, $γ$ (°) | 90, 90, 90 | 90, 90, 90 |
| Resolution range (Å) | 51.59–2.0 (2.034–2.0) | 51.47–2.0 (2.034–2.0) |
| Unique reflections | 38,319 (1881) | 38,188 (1880) |
| Multiplicity | 66.94 (25.42) | 59.92 (26.46) |
| Merged lattices | 14,357 | 13,509 |
| Completeness (%) | 99.91 (100) | 99.91 (100) |
| Mean $I$/sigma ($I$) | 3.247 (0.654) | 3.28 (0.791) |
| Wilson $B$-factor (Å²) | 35.45 | 33.22 |
| $R_{split}$ (%) | 11.1 (86.3) | 11.1 (76.5) |
| $CC_{1/2}$ | 0.989 (0.32) | 0.988 (0.39) |
| **Refinement statistics** | | |
| Resolution range used in refinement (Å) | 24.17–2.0 (2.07–2.0) | 23.91–2.0 (2.07–2.0) |
| Reflections used in refinement | 38,253 (3745) | 38,119 (3729) |
| Reflections used for $R_{free}$ | 1909 (206) | 1897 (203) |
| $R_{work}$ (%) | 15.91 (30.40) | 15.16 (28.44) |
| $R_{free}$ (%) | 19.56 (32.27) | 18.37 (31.18) |
| RMSD, bond distances (Å) | 0.007 | 0.007 |
| RMSD, bond angles (°) | 0.79 | 0.81 |
| Ramachandran favoured (%) | 99.27 | 98.79 |
| Ramachandran allowed (%) | 0.73 | 1.21 |
| Ramachandran outliers (%) | 0 | 0 |
| Rotamer outliers (%) | 0.79 | 1.05 |
| Clashscore | 1.86 | 2.58 |
| Protein residues (R2b + NrdI) | 415 (299 + 116) | 416 (298 + 118) |
| Average $B$-factor (Å²) | 47.27 | 43.66 |
| Macromolecules | 47.24 | 43.64 |
| Ligands | 34.66 | 30.38 |
| Solvent | 50.57 | 47.09 |
| Number of non-H atoms | 3655 | 3652 |
| Macromolecules | 3462 | 3477 |
| Ligands | 33 | 33 |
| Solvent | 160 | 142 |

ratio of 1:1 and set up for batch crystallization. Manganese was present both during the complex reconstitution and in the crystallization condition. Crystals for two different datasets were prepared to investigate different oxidation states of NrdI. The crystals for the first dataset were grown under aerobic conditions, yielding bright orange crystals, indicating that NrdI was fully oxidized (*Figure 2— figure supplement 1*). The structure from these crystals is later referred to as $BcR2b_{MnMn}$–NrdI$_{ox}$ (PDB ID: 7Z3D). The second dataset was obtained by reducing $BcR2b_{MnMn}$–NrdI$_{ox}$ crystals chemically with sodium dithionite in an anaerobic environment. Consequently, the crystal colour changed from bright orange to faint yellow, indicating that NrdI underwent a two-electron reduction to the hydroquinone state (*Figure 2—figure supplement 1*). The crystals were subsequently kept under anaerobic conditions, preventing reoxidation of NrdI. The corresponding structure is denoted $BcR2b_{MnMn}$–NrdI$_{hq}$ (PDB ID: 7Z3E).

## Overall structure

The two structures were solved in space group $C222_1$ and are of very similar quality with a resolution of 2.0 Å and similar unit cell dimensions (*Table 1*). The asymmetric unit contains one monomer of the 1:1 $BcR2b$–NrdI complex. The physiological dimer can be generated by applying crystal symmetry (*Figure 2A*). Clear electron density maps allowed us to model residues 1–299 of 322 for R2b and 1–116 of 118 for NrdI in $BcR2b_{MnMn}$–NrdI$_{ox}$ and 1–298 of R2b and all 118 residues of NrdI in $BcR2b_{MnMn}$–NrdI$_{hq}$. Overall $BcR2b_{MnMn}$–NrdI$_{ox}$ and $BcR2b_{MnMn}$–NrdI$_{hq}$ are similar, as indicated by the Cα root-mean-square deviation (RMSD) value of 0.16 Å (*Figure 2B*). Refinement of both structures was conducted independently from each other following the same protocol (see Materials and methods for details) with final $R_{work}/R_{free}$ of 0.16/0.20 for $BcR2b_{MnMn}$–NrdI$_{ox}$ and 0.15/0.18 for $BcR2b_{MnMn}$–NrdI$_{hq}$ (*Table 1*). Hammerstad et al. previously reported two crystal structures of the $BcR2b$–NrdI complex (*Hammerstad et al., 2014*), obtained by single-crystal X-ray diffraction at a synchrotron source and under cryogenic conditions. Here, however, R2b harbours a di-iron-active site. These structures will be referred to as $BcR2b_{FeFe}$–NrdI-1 and $BcR2b_{FeFe}$–NrdI-2 (PDB ID: 4BMO and 4BMP, respectively). Structural alignment between the published synchrotron and our XFEL structures shows that the overall fold of the complex is similar with Cα-RMSD values between 0.42 and 0.47 Å (*Figure 2B*). Compared to the

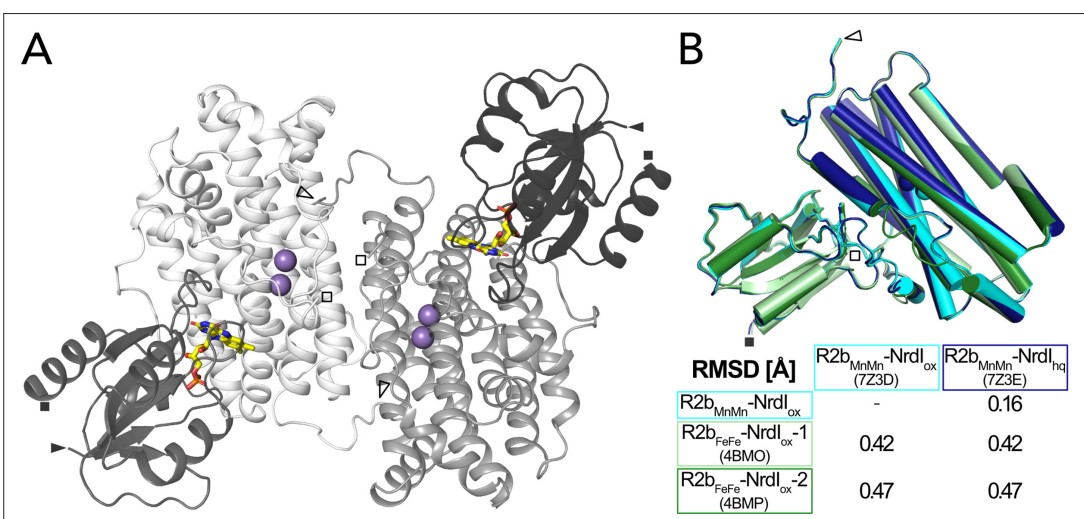

| RMSD [Å] | R2b$_{MnMn}$-NrdI$_{ox}$ (7Z3D) | R2b$_{MnMn}$-NrdI$_{hq}$ (7Z3E) |
|---|---|---|
| R2b$_{MnMn}$-NrdI$_{ox}$ | - | 0.16 |
| R2b$_{FeFe}$-NrdI$_{ox}$-1 (4BMO) | 0.42 | 0.42 |
| R2b$_{FeFe}$-NrdI$_{ox}$-2 (4BMP) | 0.47 | 0.47 |

**Figure 2.** Structure of the $BcR2b$-NrdI complex. (**A**) Structure of the $BcR2b_{MnMn}$–NrdI$_{ox}$ complex (PDB ID: 7Z3D). In each 1:1 R2b–NrdI dimer, R2b and NrdI are coloured in lighter and darker grey, respectively. The second dimer is generated by crystal symmetry. Manganese ions are represented as purple spheres and flavin mononucleotide as yellow sticks at the R2b–NrdI interface. (**B**) Superimposition of all structures of the $BcR2b$–NrdI dimer with their corresponding Cα-RMSD in the table. Individual structures are colour coded identically in the figure and the table. Visible N-termini are marked with an arrow (►) and C-termini with a square (■).

The online version of this article includes the following figure supplement(s) for figure 2:

**Figure supplement 1.** Crystallization of the di-manganese $BcR2b$–NrdI complex.

**Figure supplement 2.** Extended C-terminus of R2b in $BcR2b_{MnMn}$–NrdI$_{ox}$.

$Bc$R2b$_{FeFe}$–NrdI structures we observe a 7–8 residue extended ordered C-terminus for R2b in our structures. The C-terminus in $Bc$R2b$_{MnMn}$–NrdI$_{ox}$ continues in the same orientation as the C-terminal helix without keeping a helical conformation. It interacts with a groove between R2b and NrdI extending the R2b–NrdI-binding area by forming hydrogen bonds to two other helices of R2b and a loop in NrdI (*Figure 2—figure supplement 2*).

## R2b complex formation prevents butterfly bend of FMN in NrdI

Three redox states of FMN are physiologically relevant for NrdI: oxidized (FMN$_{ox}$), neutral semiquinone (FMN$_{sq}$), and anionic hydroquinone (FMN$_{hq}$) (*Cotruvo et al., 2013*; *Røhr et al., 2010*; *Figure 3*). Reduction of free FMN causes the isoalloxazine to bend along the N5–N10 axis leading to a 'bufferfly bend' where the isoalloxazine moiety deviates from planarity (*Figure 4*; *Zheng and Ornstein, 1996*).

NrdI binds one molecule of FMN noncovalently. The pyrimidine ring of the 7,8-dimethyl-isoalloxazine ring is buried and tightly bound to the protein by a combination of hydrogen bonds and π-stacking interactions while the benzene ring and part of the phosphate tail are solvent exposed (*Figure 4C*). The crystal structure of $Bc$NrdI has been described previously by *Røhr et al., 2010* and shows the FMN-binding pocket on the protein surface. The same study also presents quantum mechanics/molecular mechanics (QM/MM) calculations of the theoretical butterfly bend for FMN bound to NrdI. The calculated angles are shallow with −3.1° for FMN$_{ox}$ and −3.9° for FMN$_{sq}$ and more pronounced with 14.1° for FMN$_{hq}$ (*Figure 4B*). The authors investigated the influence of photoreduction on the butterfly bend in FMN bound to $Bc$NrdI. The crystals for the first structure (PDB ID: 2X2O, 1.1 Å resolution) were produced from oxidized protein. The measured butterfly bend in this structure is 4.6° and does thus not correspond well to the calculated FMN$_{ox}$ or FMN$_{sq}$ angle. It does however resemble the calculated theoretical butterfly bend of 3.3° of the physiologically not relevant anionic semiquinone state of FMN (FMN$_{sq−}$) (*Figure 4B*). The second NrdI structure (PDB ID: 2X2P, 1.2 Å resolution) structure was produced from protein in the semiquinone state and the butterfly bend was 11° after data collection, similar to the calculated FMN$_{hq}$ angle (*Figure 4B*). The authors could confirm the reducing effect of the synchrotron radiation on FMN by comparing Raman spectra of the crystals before and after data collection. These structures will be referred to as $Bc$NrdI$_{ox+e}$ and $Bc$NrdI$_{sq+e}$ to emphasize the photoreduction. Johansson et al. observed the same discrepancy between the butterfly bend for a synchrotron structure of initially oxidized NrdI from *Bacillus anthracis* ($Ba$NrdI), a protein with 99% sequence identity to $Bc$NrdI and identical FMN protein environment (*Johansson et al., 2010*). $Ba$NrdI$_{ox+e}$ (PDB ID: 2XOD, 1.0 Å resolution) is even more bent than $Bc$NrdI$_{ox+e}$ with 5.7°, indicating significant photoreduction of FMN during data collection (*Figure 4—figure supplement 1*). *Figure 4B* lists the FMN angles of the calculated and measured structures.

The use of SFX allowed us, for the first time, to investigate the conformation of oxidized FMN in NrdI. In $Bc$R2b$_{MnMn}$–NrdI$_{ox}$, the isoalloxazine moiety of FMN$_{ox}$ is almost planar, with a butterfly bend of 0.2°. This value is comparable to the angle of −3.1° for FMN$_{ox}$ in NrdI calculated by QM/MM (*Røhr et al., 2010*). Unexpectedly, the bend of FMN$_{hq}$ in the $Bc$R2b$_{MnMn}$–NrdI$_{hq}$ structure is minimal with only 0.1° and thus similar to FMN$_{ox}$ in the $Bc$R2b$_{MnMn}$–NrdI$_{ox}$ structure (*Figure 4A, B*). Additionally, we calculated an isomorphous difference (Fo(ox)–Fo(hq)) map of $Bc$R2b$_{MnMn}$–NrdI$_{ox}$ and $Bc$R2b$_{MnMn}$–NrdI$_{hq}$ using the phases of the oxidized dataset. These maps reduce model bias by directly comparing the experimental data and are very sensitive to subtle changes of atom positions in different datasets (*Rould and Carter, 2003*). The Fo(ox)–Fo(hq) map shows a slight movement of the oxygen on C4 of FMN indicating a small twist of the pyrimidine ring between both structures but no further bending (*Figure 5*). This angle of $Bc$R2b$_{MnMn}$–NrdI$_{hq}$ does not correspond to either the calculated bend of

**Figure 3.** The physiologically relevant oxidation states of flavin mononucleotide (FMN) in NrdI. The red dashed line marks the virtual axis between N5 and N10. The ribityl phosphate group is denoted as R.

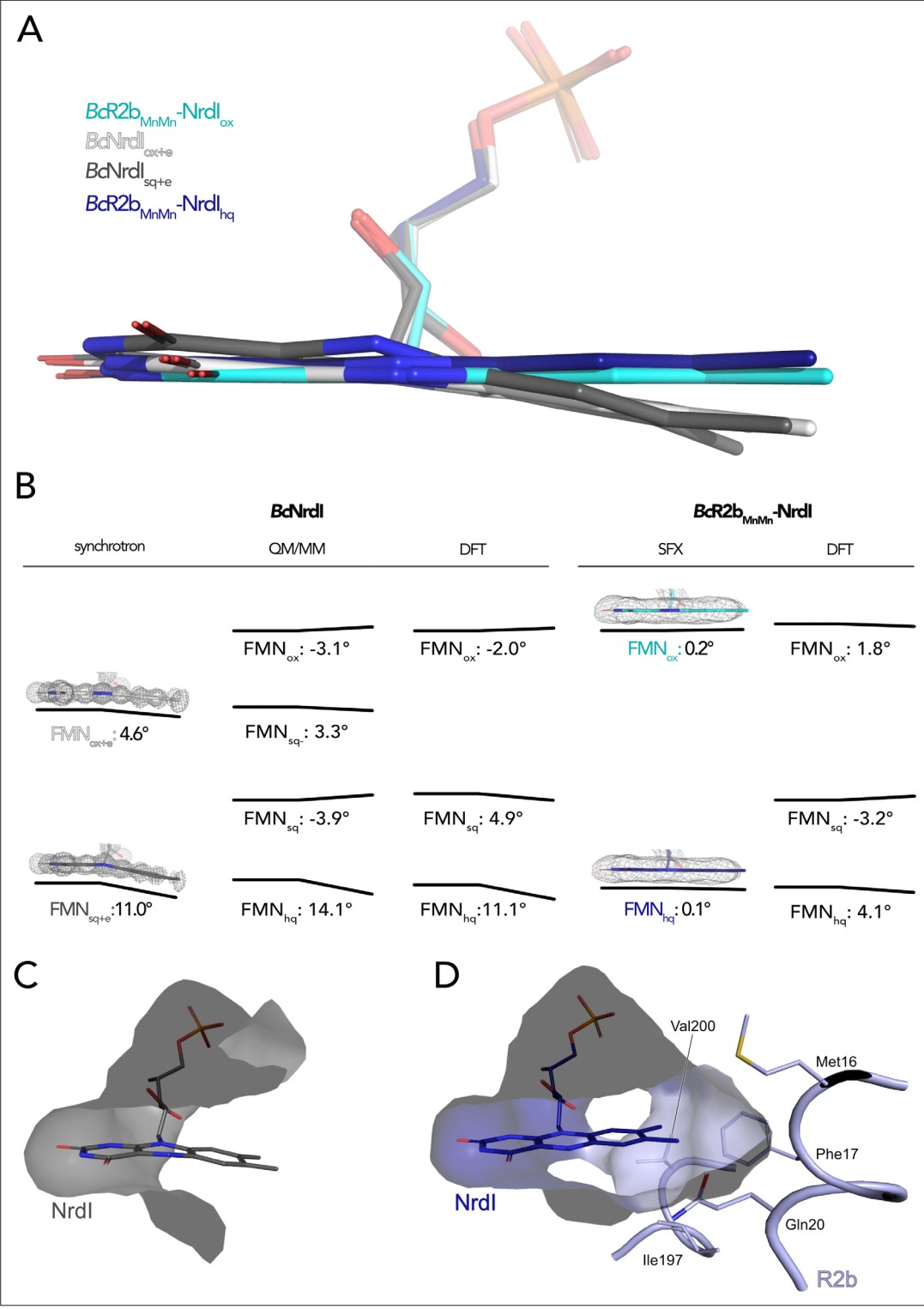

**Figure 4.** The butterfly bending conformations of flavin mononucleotide (FMN) in different redox states. (**A**) Overlay of FMN cofactors of *B. cereus* NrdI in different crystal structures: *Bc*R2b_MnMn–NrdI_hq (dark blue), *Bc*R2b_MnMn–NrdI_ox (cyan), *Bc*NrdI_sq+e (dark grey, PDB ID: 2X2P), *Bc*NrdI_ox+e (white, PDB ID: 2X2O). (**B**) Comparison of experimentally measured and computationally calculated FMN bending angles for *Bc*NrdI alone and in complex with *Bc*R2b. 2Fo–Fc maps are contoured at 2 σ for synchrotron and serial femtosecond crystallography (SFX) structures, the models used are the same as in panel A and color-coded identically. Theoretical butterfly angles obtained by quantum mechanics/molecular mechanics (QM/MM) calculations by *Røhr et al., 2010* and compared

*Figure 4 continued on next page*

*Figure 4 continued*

to angles obtained by density functional theory (DFT) calculations in this work. (**C**) The benzene ring of FMN in *Bc*NrdI$_{sq+e}$ (PDB ID: 2X2P) is solvent exposed. (**D**) In *Bc*R2b$_{MnMn}$–NrdI$_{hq}$, the FMN-binding pocket is closed by residues contributed by R2b. Relevant residues are shown in sticks. The surface of the FMN-binding pocket in NrdI in panels (C) and (D) is shown in grey.

The online version of this article includes the following figure supplement(s) for figure 4:

**Figure supplement 1.** Flavin mononucleotide (FMN) in *Bacillus anthracis* NrdI$_{ox+e}$.

**Figure supplement 2.** Density functional theory (DFT) calculations.

FMN of 14.1° nor the experimental bending angle of *Bc*NrdI$_{sq+e}$ of 11°. In the R2b–NrdI complex, the solvent exposed part of FMN is covered by R2b, which closes the binding pocket with residues Met16, Phe17, Gln20, Ile197, and Val200 (**Figure 4D**). In particular, nonpolar contacts formed between the outermost methyl groups of FMN and the side chains of Phe17 and Gln20 are sterically hindering the isoalloxazine moiety from bending. Both side chains are in turn held in place through a network of interactions involving a second shell of residues from R2b and NrdI that prevent Phe17 and Gln20 from changing conformation should FMN bend (**Supplementary file 1**).

In order to quantify the strain effects induced by R2b on FMN, we performed quantum chemical DFT calculations based on the R2b–NrdI complex structures or structures of NrdI alone (PDB IDs: 2X2O, 2X2P), including all first and second sphere interactions between FMN and the surrounding protein and water molecules, with around 170–250 atoms modelled quantum mechanically (**Figure 4—figure supplement 2**). The DFT calculations suggest that the isoalloxazine ring of FMN is bent in NrdI, with bending angles of −2° to 11° depending on the redox state, which compares well with the angles calculated at the QM/MM level by Røhr et al. (**Figure 4B**, **Supplementary file 2**). In contrast, we find that the binding of R2b to NrdI results in a more planar isoalloxazine ring for all oxidation states, with a bending angle of only −3° to +4°, which agrees with the angles obtained experimentally by SFX (**Figure 4B**) although not fully reproducing the planarity observed in the SFX crystal structures. This geometric distortion introduces molecular strain on FMN with strain energies of around 70–150 meV higher in the R2b–NrdI complex relative to NrdI alone (**Supplementary file 2**). We find that these effects tune the redox potential of FMN by 118 mV and 54 mV in the R2b–NrdI complex relative to NrdI alone for the oxidized/semiquinone (FMN$_{sq}$/FMN$_{ox}$) and semiquinone/hydroquinone (FMN$_{hq}$/FMN$_{sq}$) redox couples, respectively (**Supplementary file 3**). These rather large redox tuning effects arise from the combination of strain and electrostatic effects.

## Reorganization of FMN environment and binding position controlled by FMN redox state

The change of redox state of FMN causes movement around the cofactor in both proteins. Alignment of the NrdI backbone of the *Bc*R2b$_{MnMn}$–NrdI$_{ox}$ and the *Bc*R2b$_{MnMn}$–NrdI$_{hq}$ structures shows a clear reorganization in the 40 s loop close to the isoalloxazine ring of FMN (**Figure 5A**, **Figure 5—figure supplement 1**). The biggest change can be seen for Cα of Gly44 and a smaller shift for Cα of Thr43 and Phe45. The reorganization of the 40 s loop of NrdI after reduction is also apparent in the superposition of both structures and in the Fo(ox)–Fo(hq) map (**Figure 5B, C**). Notably, Gly44 of NrdI is flipped by 180° and forms a hydrogen bond with the hydrogen of N5 on FMN. The hydrogen bond between O4 of FMN and Gly46 shifts slightly. Thr43 in turn is shifted towards FMN and its side chain is rotated by 180°; Phe45 is also slightly rotated (**Figure 5C**). This redox-dependent conformational change of residues 43–45 was previously also observed in crystal structures of *Bc*NrdI and *Ba*NrdI in the absence of the R2b subunit (PDB ID: 2X2O, 2X2P, 2XOD, 2XOE) (**Johansson et al., 2010**; **Røhr et al., 2010**).

Redox-induced changes are also visible on the side of the isoalloxazine ring facing R2b. An unknown molecule is bound within 5 Å from the reactive C4a of FMN in *Bc*R2b$_{MnMn}$–NrdI$_{ox}$, which binds $O_2$ to reduce it to $O_2^{\bullet-}$ (**Ghisla and Massey, 1989**). The molecule is 3-coordinated to Trp74 (NrdI), Ser196 (R2b), and a water (p-ox in **Figure 5D, E**). A similar density was previously observed in the di-iron *Bc*R2b–NrdI synchrotron structures where it was modelled as a chloride ion. In our structure, placing a water in this position does not sufficiently explain the density while modelling it as a chloride ion leaves no residual density (**Figure 5—figure supplement 2**). Chloride is abundant in the crystallization condition; however, serine and tryptophan are not typical chloride ligands (**Carugo, 2014**).

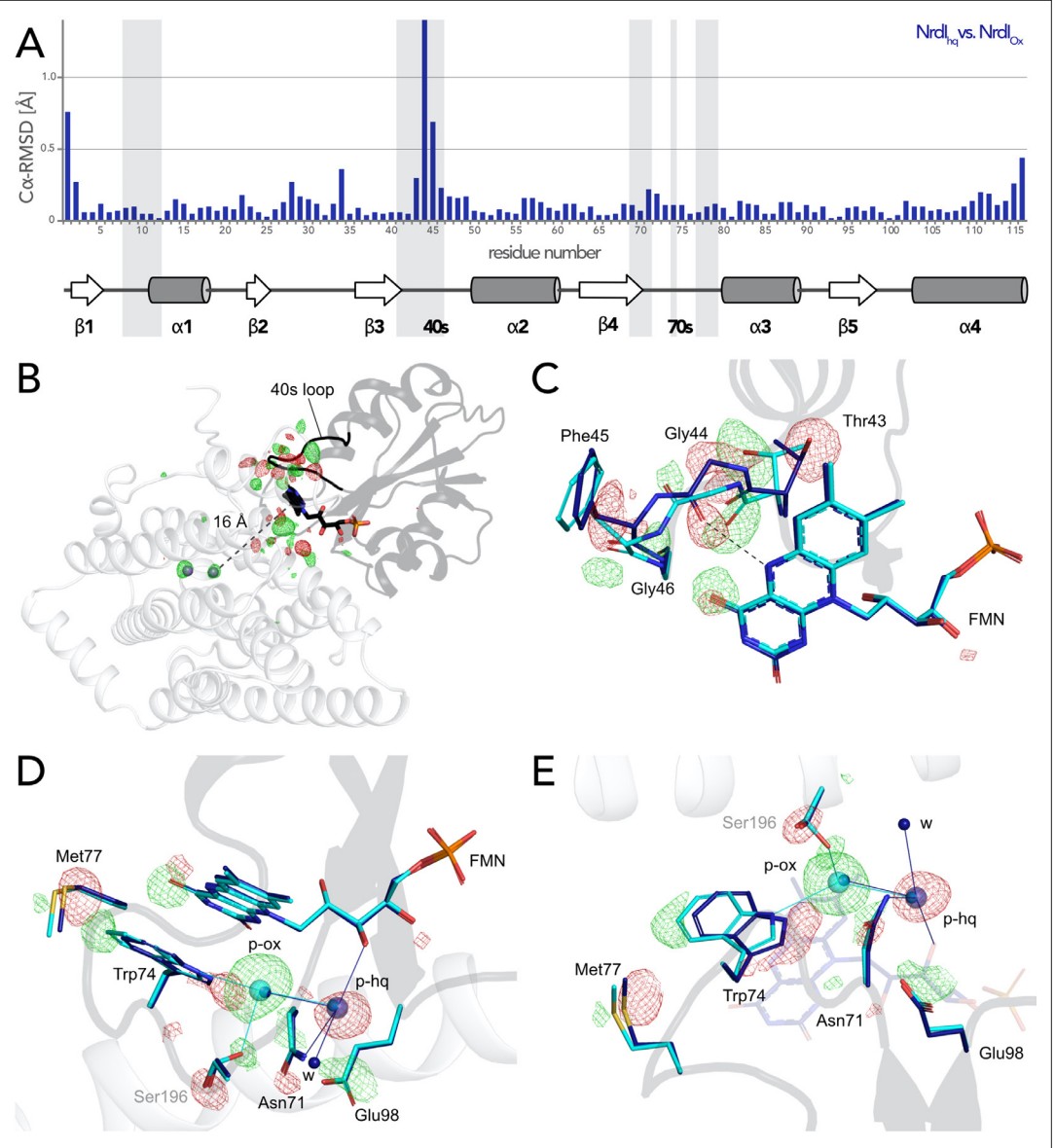

**Figure 5.** Flavin mononucleotide (FMN) environment in the oxidized and hydroquinone *Bc*R2b–NrdI complex. (**A**) Structural alignment of NrdI in *Bc*R2b$_{MnMn}$–NrdI$_{ox}$ compared to *Bc*R2b$_{MnMn}$–NrdI$_{hq}$. Cα-RMSD in Å are shown for each residue of NrdI between the two models. The secondary structure assignment corresponding to the residues of NrdI is represented in the cartoon below. NrdI interactions with FMN are marked with light grey background (**B**) Fo(ox)–Fo(hq) map contoured at 4.5 σ for the *Bc*R2b–NrdI complex with positive density in green and negative density in red. FMNs are shown in sticks, manganese ions as purple spheres, R2b as transparent cartoon in white and NrdI in transparent black. Differences between both datasets cluster around the FMN at the R2b/NrdI interface. The distance between metal site and FMN is around 16 Å, marked with a dashed line. (C–E) Closeups of the difference density around the FMN from different angles including superposition of *Bc*R2b$_{MnMn}$–NrdI$_{hq}$ in dark blue and *Bc*R2b$_{MnMn}$–NrdI$_{ox}$ in cyan. Fo(ox)–Fo(hq) map contoured at 4.5 σ for all panels. Adjacent secondary structure elements are shown as transparent cartoon in white for R2b and black for NrdI. (**C**) Rearrangement of the 40 s loop of NrdI. Gly44 flips 180° in the hydroquinone state and forms a hydrogen bond with N5 of FMN (shown as dashed line). (**D**) Rearrangements of side chains on the R2b facing side of FMN. Residues of NrdI are labelled in black, Ser196 from R2b in light grey. An unexplained density, larger than water, moves between the structures, named p-ox and p-hq and is marked as transparent big spheres; waters are represented as small opaque spheres. (**E**) Same side chains as in (D) are shown at a different angle facing the FMN. FMN is shown transparent in the background for clarity.

The online version of this article includes the following figure supplement(s) for figure 5:

*Figure 5 continued on next page*

*Figure 5 continued*

**Figure supplement 1.** Electron density around the 40 s loop of NrdI in the oxidized and reduced complex structures.

**Figure supplement 2.** Different modelling results for unidentified density close to flavin mononucleotide (FMN).

**Figure supplement 3.** Channel between R2b and NrdI.

The exact nature of the molecule could not be determined and was thus designated as an unknown atom in the final model. Interestingly, the binding position undergoes a redox-dependent switch: In the reduced structure the density is instead found at a position between the oxygen on C2 of $FMN_{hq}$ and Asn71 of NrdI, 7 Å from the reactive C4a. Two adjacent water molecules make the unknown molecule 4-coordinated (p-hq in *Figure 5D, E*). To accommodate for the change of position residues Met77, Trp74, Asn71, and Glu98 from NrdI and Ser196 from R2b move in a plane parallel to the FMN isoalloxazine ring (*Figure 5D, E*). The unidentified molecule exchanges position with two different water molecules of the well-ordered solvent network connecting FMN with the active site in R2b. The solvent network is housed by a channel between both proteins lined by the side chains of Ser162, Tyr166, Lys263, and Asn267 and the mainchain of Glu195 and Ser196 (*Hammerstad et al., 2014*). The position of the density in $BcR2b_{MnMn}$–$NrdI_{ox}$ is close to the channel entrance on the R2b surface marked by Lys263, Asn267, and Ser196 (*Figure 5—figure supplement 3*). Despite the large structural rearrangements around the FMN, the Fo(ox)–Fo(hq) map clearly shows that no redox-induced movement protrudes further down the channel and that the coordination of the metal site is unaffected by the change of NrdI redox state (*Figure 5B* and *Figure 5—figure supplement 3*).

## Access to the metal site is not gated by FMN oxidation state, complex formation, or manganese binding

Both SFX BcR2b–NrdI structures display a di-manganese metal centre with similar coordination. In $BcR2b_{MnMn}$–$NrdI_{ox}$, the two metal ions of R2b are refined at full occupancy and the coordination sphere is clearly defined in the electron density map (*Figure 6A*). Mn1 is 4-coordinated by His96, Glu93, Asp62, and Glu195. Mn2 is 5-coordinated by His198, Glu93, Glu195, and Glu161. Notably, Glu195 and Glu161 exhibit two alternative conformations each and alternately coordinate Mn2 in a monodentate or bidentate fashion. Importantly, while Glu195 bridges the two ions, Glu161 coordinates only Mn2. No coordinating waters could be identified (*Figure 6B*). The protein complex used in the crystallization for the $BcR2b_{MnMn}$–$NrdI_{ox}$ dataset has not undergone any oxidation of the metal site and both manganese ions are in the Mn(II)/Mn(II) oxidation state (see Materials and methods for details). Notably, the reduction of FMN within NrdI occurs at about 16 Å from the metal site and does not affect the metal coordination in BcR2b–NrdI (*Figure 5A*). The Fo(ox)–Fo(hq) map shows slightly lower occupancy for both manganese ions in $BcR2b_{MnMn}$–$NrdI_{hq}$ (*Figure 6—figure supplement 1*). The loss of metals can be explained by the reduction treatment of the crystals for this dataset and Mn1 was modelled with 80%, Mn2 with 90% occupancy in the reduced structure. Both structures show the same metal–metal distance of 3.8–3.9 Å within experimental error for this resolution (*Figure 6B*).

The metal–ligand Glu161 is likely a key player in the activation of the di-manganese centre as it is proposed to gate the access to the metal site for the oxidant produced by NrdI (*Boal et al., 2010*). Two main conformations have been observed for this ligand: In the 'closed' conformation the glutamate interacts only with Mn2 and is proposed to prevent $O_2^{\bullet-}$ from reaching the metal site by obstructing the channel. In the 'open' conformation, on the other hand, Glu161 bridges both manganese ions leaving space for three water molecules to connect Mn2 to the water network in the channel between FMN and the metal site. The open conformation of the equivalent glutamate has been observed in structures of di-manganese R2b alone, for example from *B. cereus*, *E. coli*, and *Streptococcus sanguinis* (PDB ID: 4BMU, 3N37, 4N83) (*Figure 6C*) but also in the R2b–NrdI complex of *E. coli* (PDB ID: 3N3A) (*Figure 6D*; *Boal et al., 2010*; *Hammerstad et al., 2014*; *Makhlynets et al., 2014*). Thus, NrdI binding to R2b is not responsible for triggering the open conformation of Glu161 (Glu158 in *E. coli* and Glu157 in *S. sanguinis*). It has been hypothesized that the presence of manganese ions in the active site could cause the glutamate to shift to the open conformation as the closed conformation is typically observed in R2b structures containing a di-iron site, for example in the $BcR2b_{FeFe}$–NrdI complex (PDB ID: 4BMP) (*Figure 6E*; *Hammerstad et al., 2014*). Indeed, since the di-iron form of R2b

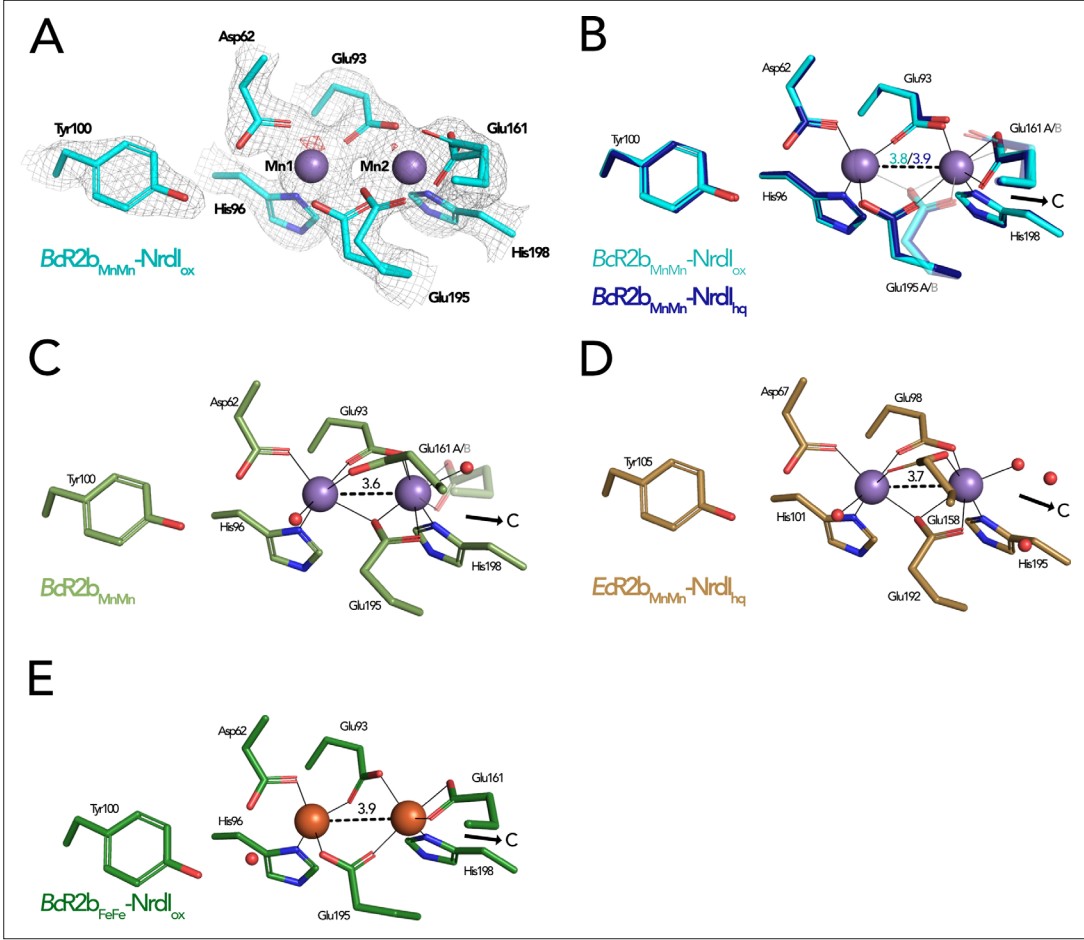

**Figure 6.** Comparison of active site architecture in different R2b structures including the radical harbouring tyrosine. Manganese ions are shown as purple, iron ions as orange, and waters as red spheres. The direction of the connecting channel between R2b and NrdI is indicated by a black arrow and 'C'. (**A**) $Bc$R2b$_{MnMn}$–NrdI$_{ox}$ active site. The 2Fo–Fc map is contoured at 2 σ shown as grey mesh and Fo–Fc is contoured at 4 σ with negative density in red (no positive density is present). (**B**) Superposition of $Bc$R2b$_{MnMn}$–NrdI$_{hq}$ and $Bc$R2b$_{MnMn}$–NrdI$_{ox}$. The metal coordination is identical for both structures. (**C**) Active site of $Bc$R2b$_{MnMn}$ (PDB ID: 4BMU). The metal site of chain A is shown with Glu161 in both the open (opaque) and closed (transparent) conformation. (**D**) Active site of $Ec$R2b$_{MnMn}$–NrdI$_{hq}$ (PDB ID: 3N3A). Glu158 is in the open conformation and two waters are in the position of the Glu161 of $Bc$R2b. (**E**) Active site of the di-iron $Bc$R2b–NrdI complex (PDB ID: 4BMP). The metal coordination is similar to the manganese containing structures with Glu161 in the closed conformation. An additional water is hydrogen bonded to Asp62 and Tyr100 but is not in interaction with iron. Metal–metal distances are shown as dashed and metal ligands as solid lines; residues in alternative conformations are shown transparent for panels (B)–(E).

The online version of this article includes the following figure supplement(s) for figure 6:

**Figure supplement 1.** Comparison of the active site in di-manganese $Bc$R2b$_{MnMn}$–NrdI structures.

---

is oxidized by $O_2$ via a NrdI-independent pathway, the movement of the glutamate is not required for radical generation. However, our SFX $Bc$R2b$_{MnMn}$–NrdI structures harbour a di-manganese site and exhibit a Glu161 in the closed conformation, thereby demonstrating that manganese in the active site is not sufficient to induce the glutamate shift. Finally, our radiation damage free structures also show that the opening of the channel towards the metal site is independent from the FMN oxidation state, as Glu161 is in the closed conformation when di-manganese R2b is in complex with either NrdI$_{ox}$ or NrdI$_{hq}$ (**Figure 6B**). Altogether, our data show that the glutamate shift of Glu161 is not caused by the presence of manganese ions, R2b–NrdI complex formation, a specific NrdI redox state or a combination of these factors.

## Discussion

In this study, we present two structures of the manganese containing R2b–NrdI complex of *B. cereus* with NrdI in two different oxidation states. These structures were obtained by room temperature SFX data collection using XFEL radiation in contrast to all previously studied R2b or NrdI structures, which were obtained by classical single-crystal synchrotron data collection at cryogenic temperatures. Determination of the oxidation state of redox-active enzymes via synchrotron radiation is problematic because exposure to X-rays exerts photoreduction on redox centres. Investigation of artefact-free oxidized enzymes is therefore exceedingly challenging with synchrotron radiation. In the context of class Ib RNRs this affects both the redox-active metal centre of R2b and FMN cofactor of NrdI. SFX eliminates the effects of photoreduction observed during synchrotron-based data collection. We describe SFX structures of the R2b–NrdI complex with NrdI in the oxidized and hydroquinone oxidation state. Both 2.0 Å structures are of similar quality as the previously published synchrotron structures of the same complex with a di-iron centre (*Hammerstad et al., 2014*) showing that the change of methodology does not affect the quality. This allowed detailed examination of structural reorganization induced by changes in FMN redox state. Notably, the FMN conformation per se changes little between the oxidized and hydroquinone oxidation state in R2b–NrdI. This is surprising since it was previously shown that FMN is bent in reduced NrdI when not bound to R2b. The complexation of R2b and NrdI thus prevents FMN bending and exerts strain on the isoalloxazine ring in the hydroquinone state, which was also reproduced in DFT calculations. Interestingly, the calculations revealed that the oxidized FMN is also under strain in the R2b–NrdI complex in comparison to NrdI alone. It could also be confirmed by the calculations that R2b causes a shift in redox potential for both redox couples ($FMN_{sq}/FMN_{ox}$ and $FMN_{hq}/FMN_{sq}$). Other flavoproteins have been shown to tune the redox potential of FMN by forcing it into a specific binding angle (*Senda et al., 2009*; *Walsh and Miller, 2003*). A recent study by Sorigué et al. presented an SFX structure of the flavoenzyme fatty acid photodecarboxylase (FAP) in complex with an oxidized flavin cofactor exhibiting a butterfly bending angle of 14° (*Sorigué et al., 2021*). Using time resolved SFX and complementary approaches, the study showed that during the enzymatic cycle of FAP the conformation of the flavin retains the butterfly bend in the different redox states. The authors conclude that the butterfly bend is enforced by the protein scaffold to promote flavin reduction. Following the same line of reasoning we propose that R2b binding to NrdI restricts FMN bending and thus, opposite to FAP, changes the FMN redox potential of the hydroquinone to favour FMN oxidation and reduction of molecular oxygen to superoxide. This notion is supported by a study by Cotruvo et al. demonstrating that superoxide production by the *in vitro* R2b–NrdI$_{hq}$ complex of *Bacillus subtilis* is about 40 times faster than the production of superoxide by free NrdI$_{hq}$ under the same experimental conditions (*Cotruvo et al., 2013*). Our calculations support that the effect of strain on the flavin can contribute to downshift the FMN redox potential, which would thermodynamically favour electron transfer. Interestingly, however, the electrostatic effects in the R2b–NrdI complex are predicted to counteract this effect. From a kinetic perspective, on the other hand, the strain imposed in the FMN$_{hq}$ state by the R2b–NrdI complex could lower the reorganization energy of electron transfer and thus kinetically favour superoxide generation. The full effects of R2b–NrdI complex formation on the FMN redox properties and their physico-chemical background are complex and warrant further investigation.

Controlling the superoxide production by complex formation could serve as a mechanism to protect the cell from production of superoxide by free NrdI. The binding of R2b may also facilitate access of $O_2$ to FMN by forming a hydrophobic binding pocket around the benzene ring of FMN. $O_2$ is reduced to superoxide at the C4a atom of the isoalloxazine ring of FMN and we observe a molecule bigger than a water bound about 5 Å away from the C4a atom in the oxidized R2b–NrdI complex. From the experimental setup it seems unlikely that the unidentified molecule is a superoxide ion because FMN never underwent a redox cycle in this structure. With a concentration of 100 mM in the crystallization condition chloride is a more likely candidate for the molecule, as also suggested by model refinement (*Figure 5—figure supplement 2*). However, refinement with a superoxide ion leaves little residual density, so the binding could fit a molecule of its size (*Figure 5—figure supplement 2*). Given the similar negative charge and the comparable electron density for the superoxide and chloride anions, the observed density could thus represent a potential binding position of superoxide after its generation. In addition, the density undergoes a redox-dependent switch of position moving further away from the reactive C4a (7 Å) towards the channel entrance in the reduced R2b–NrdI complex. This move

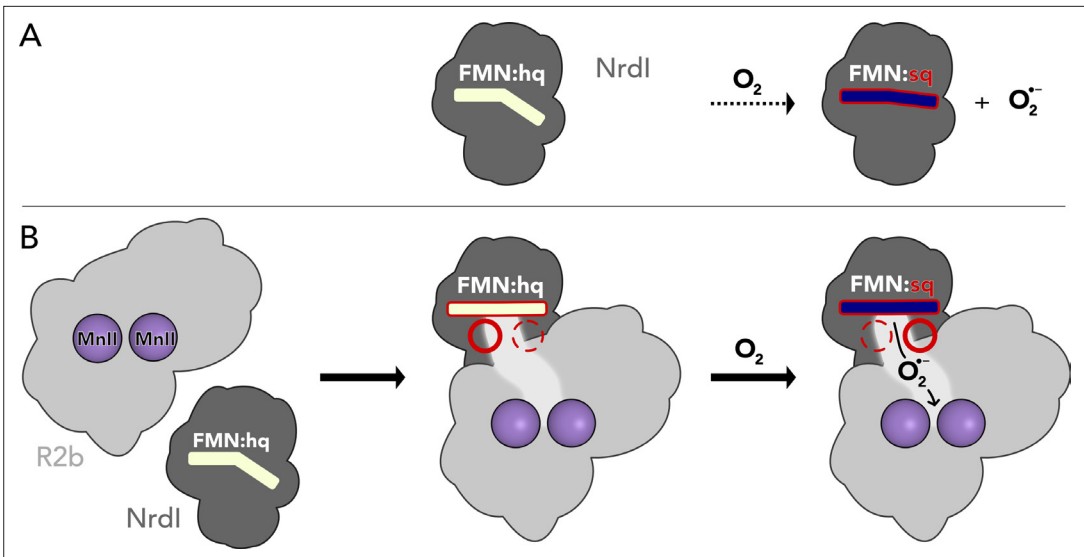

**Figure 7.** Proposed mechanism for the assembly of the R2b–NrdI complex. (**A**) Free NrdI$_{hq}$ and molecular oxygen react slowly to produce superoxide and NrdI$_{sq}$. (**B**) R2b and NrdI form a complex which imposes strain on the flavin mononucleotide (FMN) bend and tunes its redox properties to favour oxygen reduction. The complex formation also generates a redox state-controlled potential superoxide-binding site close to the reactive carbon of FMN, indicated by the dashed and solid circles. Upon exposure to molecular oxygen, superoxide is generated in the complex and shuttled towards R2b.

is accompanied by a change of the putative coordination sphere. Importantly, in both oxidized and reduced states, the unknown molecule is integrated into the conserved hydrogen-bonding network connecting FMN and the metal site and switches position with a water. Even though the exact nature of the observed electron density is unknown we conclude that the binding properties of these key positions are controlled by the FMN redox state and could mark the route superoxide would travel after being produced by the FMN.

Taken together, our results suggest that the production of superoxide by NrdI and the radical generation in R2b is elegantly orchestrated by the formation of the R2b–NrdI complex as well as redox state control of binding positions (*Figure 7*). R2b–NrdI binding induces conformational strain in the flavin as well as other effects influencing its redox properties to promote superoxide generation. The interaction surface also provides binding positions that are controlled by the redox state of the flavin, presumably involved in gating of channel and metal site access.

## Materials and methods
### Protein expression
The plasmids containing the genes for both *Bc*R2b (pET22b-Bc*r2b*) and *Bc*NrdI (pET22b-Bc*nrdI*) were kindly provided by Marta Hammerstad and Kristoffer Andersson (University of Oslo). The protein expression of *Bc*NrdI was adapted from *Røhr et al., 2010*. *E. coli* BL21(DE3) (New England Biolabs, Frankfurt am Main, Germany) cells were transformed with the pET22b-Bc*nrdI* plasmid. A preculture of Lysogeny Broth medium (Formedium, Norfolk, UK) was inoculated with a single colony, grown over night at 37°C and 200 rpm. The next day large-scale cultures of 1.6 l Terrific Broth medium (Formedium) per glass bottle supplemented with 100 µg/ml carbenicillin (Alfa Aesar, Kandel, Germany) and 1:10,000 (vol/vol) antifoam 204 (Merck, Darmstadt, Germany) were inoculated with 0.5% (vol/vol) of the preculture. The cells were incubated in a LEX bioreactor (Epiphyte3, Toronto, Canada) at 37°C until an optical density at 600 nm of about 0.8 was reached. The cultures were cooled down to 20°C, and the protein expression was induced with 0.8 mM isopropyl β-D-1-thiogalactopyranoside (IPTG) (Formedium). The cells, harvested after 12–16 hr of expression by centrifugation at 4000 × *g* for 20 min, formed a dark grey pellet. A volume of 1 l culture formed 12–19 g wet cell pellet. Expression of *Bc*R2b was adapted from *Tomter et al., 2008* and the protein was expressed in a similar way as

BcNrdI. To express BcR2b metal-free, ethylenediaminetetraacetic acid (EDTA) (PanReac AppliChem, Darmstadt, Germany) was added to the large-scale cultures shortly before induction with IPTG to a final concentration of 1 mM. A volume of 1 l culture gave about 8 g wet cell pellet. The pellets were flash frozen in liquid nitrogen and stored at −20°C until further use.

## Purification

### BcNrdI

The purification of NrdI was adapted from *Røhr et al., 2010*. About 20 g of bacterial cell pellet was resuspended in lysis buffer (100 mM Tris–HCl pH 7.5) supplemented with a tablet of EDTA-free cOmplete Protease Inhibitor S2 Cocktail Tablet (PIC) (Roche, Solna, Sweden) and DNase (PanReac Applichem) was added. Cells were lysed with a sonicator, Sonics VCX130 (Sonics, Newtown, CT), and the soluble fraction separated from cell debris by centrifugation at 40,000 × *g* for 30 min at 4°C. The NrdI protein was precipitated by slow addition of ammonium sulphate ($NH_4SO_4$) to the lysate to a final concentration of 60% (wt/vol) (0.37 g/ml) while stirring at 4°C. The precipitate was pelleted by centrifugation at 20,000 × *g* for 20 min and 4°C and subsequently solubilised with a minimal volume of size-exclusion chromatography (SEC) buffer (50 mM Tris pH 7.5). The protein was desalted by dialysis overnight at 4°C against dialysis buffer (10 mM Tris pH 7.5). The desalted protein was filtered through a 0.45 µm filter to remove precipitate and loaded onto a Q Sepharose High Performance 5 ml anion exchange column (Cytiva, Uppsala, Sweden), washed with SEC buffer and eluted with a gradient from 0% to 50% elution buffer (50 mM Tris–HCl pH 7.5, 1 M KCl). The fractions containing the desired protein could be identified by the bright orange colour of NrdI (*Figure 2—figure supplement 1*). Relevant fractions were pooled, concentrated with a Vivaspin 20 centrifugal concentrator with a 30,000 Da molecular weight cut-off polyethersulfone membrane (Vivaspin 30 k concentrator) (Sartorius, Göttingen, Germany) and injected onto a HiLoad Superdex 75 prep grade size-exclusion column (Cytiva). Orange fractions were analysed for purity by sodium dodecyl sulphate–polyacrylamide gel electrophoresis (SDS–PAGE) and fractions with a 95% or higher purity were pooled. The theoretical molecular weight of NrdI was calculated to be 13,449 Da with ProtParam (*Gasteiger et al., 2005*) and the extinction coefficient $\varepsilon_{447} = 10.8$ mM$^{-1}$ cm$^{-1}$ (*Berggren et al., 2014*) was used to determine the protein concentration using UV–vis spectroscopy. The protein was concentrated to 25 mg/ml, flash frozen in liquid nitrogen and stored at −80°C until further use.

### BcR2b

The R2b purification protocol was adapted from *Tomter et al., 2008*. The protein was produced metal free; for that purpose, EDTA was included in the lysis buffer to inhibit metal uptake. About 20 g of bacterial cell pellet was resuspended in lysis buffer (100 mM Tris–HCl pH 7.5, 1 mM EDTA) with a tablet of PIC and lysed by sonication. Streptomycin sulphate was added to a total of 2.5% (wt/vol) and incubated for 10 min at 20°C for DNA precipitation. The crude lysate was cleared by centrifugation for 30 min at 40,000 × *g*. The supernatant was cleared from contaminants by adding 40% (0.24 g/ml) $NH_4SO_4$ at 20°C followed by centrifugation at 20,000 × *g* for 20 min. The $NH_4SO_4$ concentration for the remaining supernatant was increased to 50% (0.31 g/ml) at 20° to precipitate R2b. After the second centrifugation at 20,000 × *g* for 20 min the pellet was dissolved with a minimal volume of lysis buffer. The protein was diluted 4× with buffer A (50 mM Tris pH 7.5, 1.5 M $NH_4SO_4$) to increase the salt concentration sufficiently. The diluted protein was filtered through a 0.45-µm membrane filter and loaded onto a column packed with 20 ml Phenyl Sepharose High Performance resin (Cytiva). The protein was washed with buffer A and eluted with a gradient with SEC buffer (50 mM Tris pH 7.5). Several column volumes were needed to elute the protein. The elution fractions were analysed by SDS–PAGE. Protein containing fractions were pooled, concentrated by a Vivaspin 50 k concentrator and injected onto a HiLoad Superdex 200 prep grade column (Cytiva). Elution fractions were analysed by SDS–PAGE and pure fractions pooled. The theoretical molecular weight and the extinction coefficient of R2b were calculated with ProtParam to be 37,017 Da and $\varepsilon_{280} = 48.36$ mM$^{-1}$ cm$^{-1}$, respectively (*Gasteiger et al., 2005*). R2b was concentrated to 50 mg/ml with a Vivaspin 50 kDa concentrator, flash frozen and stored at −80°C until further use. A total of 20 g bacterial cell pellet yielded about 100–150 mg pure protein.

## Quantification of R2b metal content with TXRF

To control the metal content of $Bc$R2b the protein was measured with TXRF spectroscopy using a Bruker PicoFox S2 spectrometer (Bruker, Billerica, MA). Three independently prepared replicates of the concentrated protein at 1.7 mM were mixed 1:1 with a gallium standard at 20 mg/l and dried on top of a siliconized quartz sample carrier. The discs were individually measured before use to avoid external contamination of the sample. The results were analysed with the Bruker Spectra software version 7.8.2.0 provided with the instrument.

## Crystallization of the $Bc$R2b–NrdI complex

The $Bc$R2b–NrdI protein complex was prepared for crystallization by first metal loading $Bc$R2b with manganese(II), yielding the reduced R2b$_{MnMn}$ and then forming the complex by adding oxidized NrdI: 0.25 mM $Bc$R2b in 50 mM Tris–HCl pH 7.5 was incubated with 12 molar equivalents of Mn(II)Cl$_2$ for 10 min at 20° C, then 0.25 mM $Bc$NrdI was added and the mixture incubated for another 15 min at 20°C to ensure complex formation. New crystallization conditions were screened since the crystallization condition published in *Hammerstad et al., 2014* could not be reproduced. Initial hits in conditions C5 of the JSCG+ crystallisation screen (0.1 M sodium HEPES pH 7.5, 0.8 M sodium phosphate monobasic monohydrate, 0.8 M potassium phosphate monobasic) and the A7 condition of the PACT premier crystallization screen (0.1 M sodium acetate pH 5.0, 0.2 M NaCl, 20% (wt/vol) PEG 6000) (both Molecular Dimensions, Sheffield, UK) were further optimized to yield crystals between 10 and 50 µm in the longest axis (*Figure 2—figure supplement 1*). The final crystallization protocol was established as described: Crystals spontaneously grew in a hanging drop vapour diffusion experiment after one to two days at 20°C. Crystallization condition A (0.1 M HEPES pH 7.0, 0.6–0.85 M sodium phosphate monobasic monohydrate, 0.6–0.85 M potassium phosphate monobasic) was manually mixed with the protein complex solution in a ratio of 1:1 (1 + 1 µl). The crystals grown in this experiment were used to produce microseeds for crystallization with the batch method. Two crystal containing drops were transferred into 50 µl of crystallization condition A in the bottom of a 1.5 ml microcentrifuge tube. A seed bead (Saint Gobain, Aachen, Germany) was placed into the solution, the crystal crushed by vigorous shaking and the mixture used as seed stock for the following crystallization experiments. Batch crystallization was set up in PCR tubes at 20°C. A volume of 40 µl of crystallization condition B (0.1 M Mn(II)Cl$_2$, 0.1 M sodium acetate 5.0, 5% PEG 6000) were pipetted on top of 40 µl of protein complex solution. A volume of 8 µl of microseed solution was added to the tube and everything mixed by pipetting. Orange rhombohedron shaped crystals typically sized between 20 and 50 µm in the longest axis grew over night, forming an orange pellet at the bottom of the tubes. The crystals were resuspended and pooled in 2-ml microcentrifuge tubes (*Figure 2—figure supplement 1*). Crystals used to collect the dataset for $Bc$R2b$_{MnMn}$–NrdI$_{ox}$ could be directly loaded into a Gastight SampleLock (Hamilton, Bonaduz, Switzerland) syringe for sample delivery. The crystals used for the $Bc$R2b$_{MnMn}$–NrdI$_{hq}$ dataset had to be chemically reduced first as described below.

## Reduction protocol

Crystals for the $Bc$R2b$_{MnMn}$–NrdI$_{hq}$ dataset were chemically reduced before data collection. The reduction affects only FMN in NrdI, since the active site already contains Mn(II)/Mn(II) due to metal loading with Mn(II)Cl$_2$. All following steps were conducted in an anaerobic glove box with O$_2$ below 10 ppm at room temperature. A volume of 900 µl of pooled crystal slurry was gently centrifuged, forming a dense crystal pellet. Of the supernatant 800 µl were carefully removed, collected separately, and supplemented with freshly prepared, anaerobic sodium dithionite (DT) to a final concentration of 20 mM. The DT-containing supernatant was gently mixed with the crystal pellet and the colour change of the crystals from a bright orange to a faint, light yellow was observed in a matter of minutes. The DT was subsequently washed out by gently spinning down the crystals, removing the supernatant without disturbing the crystal pellet, replacing it with anaerobic wash buffer (100 mM MnCl$_2$, 50 mM sodium acetate pH 5.0, 2.5% (wt/vol) PEG 6000, 10% (vol/vol) glycerol) and gently resuspending the crystals. This washing step was repeated three times, adding wash buffer in the last step to a volume of 800 µl. The final concentration of DT in the crystal slurry was below 4 µM. All of the crystal slurry was loaded into a 1 ml gastight SampleLock Hamilton syringe and stored in the anaerobic box until sample injection.

## Data collection

The $Bc$R2b–NrdI crystals were initially tested for stability and diffraction quality at SACLA in Japan during experiment 2017B8085. The sample was delivered with a grease extruder setup installed on site (**Sugahara et al., 2015**; **Tono et al., 2013**). Hydroxyethyl cellulose matrix (**Sugahara et al., 2017**) and the protein crystals were mixed together in a volume ratio matrix to crystal pellet of 9:1 and the mixture ejected with a HPLC pump through a 150 µm nozzle with a flow rate of 1–1.5 µl/min, delivering the sample into the XFEL beam. X-rays were delivered as <10 fs long pulses at 10.9 keV with 30 Hz repetition rate and a typical pulse energy of around 0.32 mJ with a beam size of $2 \times 2$ µm$^2$ (fwhm). The diffraction of the sample was recorded 100 mm downstream of the interaction point on an Octal MPCCD detector. The initial data collection showed stability of the crystals over days and a diffraction quality to about 2 Å.

The datasets used in this paper were collected at LCLS, California during experiment LU50. X-ray pulses at 9.5 keV with a pulse energy of 4 mJ, 30 Hz repetition rate and a duration of around 35 fs were generated and used for X-ray diffraction in the macromolecular femtosecond crystallography (MFX) experimental hutch (**Sierra et al., 2019**). The diffraction was recorded on a Rayonix MX340 (Rayonix L.L.C, Evanston, USA) detector. The sample was delivered into the X-ray interaction point with the drop-on-tape method; the detailed method was described by **Fuller et al., 2017**. Briefly, crystal slurry in a 1-ml SampleLock syringe was pumped with a syringe pump (KD scientific, Holliston, MA) at a flow rate of 8–9 µl/min through a silica capillary into a 6 µl sample reservoir. An acoustic transducer transferred crystal containing droplets of 2.5–4 nl volume onto a Kapton tape, which transported the droplets into the X-ray beam at 28°C and 27% relative humidity with a speed of 300 mm/s, which resulted in the crystals being exposed to the He environment for about 0.8 s. The enclosure of the setup was filled with a He atmosphere with an $O_2$ level below 0.1% for the $Bc$R2b$_{MnMn}$–NrdI$_{hq}$ data collection.

## Processing, structure determination, and refinement

The datasets for $Bc$R2b$_{MnMn}$–NrdI$_{hq}$ and $Bc$R2b$_{MnMn}$–NrdI$_{ox}$ were processed with cctbx.xfel (**Brewster et al., 2019b**; **Hattne et al., 2014**; **Sauter, 2015**) and DIALS (**Brewster et al., 2018**; **Winter et al., 2018**). We performed joint refinement of the crystal models against the detector position for each batch to account for small time-dependent variations in detector position and also corrected for the Kapton tape shadow (**Fuller et al., 2017**). Data were scaled and merged to 2.0 Å resolution using cctbx.xfel.merge with errors determined by the ev11 method (**Brewster et al., 2019a**). The final resolution cut-off was based on values of CC$_{1/2}$ (where it does not monotonically decrease as a function of resolution anymore) and multiplicity (where it falls off the 10-fold threshold), as well as on $R$-factors after initial refinement. Data statistics are available in **Table 1**. Both structures were solved by molecular replacement and refined independently with the PHENIX Suite (**Liebschner et al., 2019**). The phases were solved with phenix.phaser (**McCoy et al., 2007**). A $Bc$R2b–NrdI complex structure (PDB ID: 4BMO; **Hammerstad et al., 2014**) was modified by manually removing waters and alternate conformations and used as a starting model. The suggested solutions were in the same space group ($C222_1$) as the starting model with similar unit cell dimensions (**Table 1**) with one complex in the asymmetric unit. The $R_{free}$ set of the 4BMO model corresponding to 5% of reflections was assigned to all datasets with phenix.reflection_tools. Restraints of the entry 'FMN' of the ligand database provided by phenix were used for the oxidized FMN in $Bc$R2b$_{MnMn}$–NrdI$_{ox}$. Restraints for the hydroquinone FMN in $Bc$R2b$_{MnMn}$–NrdI$_{hq}$ were generated with phenix eLBOW (**Moriarty et al., 2009**). The datasets were iteratively refined with phenix.refine (**Afonine et al., 2012**), examined and built in coot (**Emsley et al., 2010**) and validated with MolProbity (**Williams et al., 2018**). Refinement of all atoms included isotropic $B$-factors, TLS parameters, occupancy and reciprocal space refinement with a high-resolution cut-off of 2.0 Å. Waters were initially added using phenix.refine and in later refinements corrected manually. The metal occupancy for the reduced structure was fixed manually after several cycles of refinement. The unknown species found in the vicinity of the FMN was modelled as unknown atom (PDB ligand ID: UNX) in the two structures. An overview over refinement and model quality statistics can be found in **Table 1**, created with phenix.table_one. The refined structures were compared and RMSD values calculated with SSM superimpose (**Krissinel and Henrick, 2004**). The contacts of FMN in both structures listed in **Supplementary file 1** were analysed with the help of LIGPLOT (**Wallace et al., 1995**).

## DFT calculations

Quantum chemical DFT models of the FMN-binding site in R2b–NrdI and NrdI systems were built based on the coordinates of the crystal structures of R2b–NrdI$_{ox}$ and R2b–NrdI$_{hq}$ (this study), and NrdI$_{ox}$ (PDB ID: 2X2O) and NrdI$_{sq}$ (PDB ID: 2X2P) (*Røhr et al., 2010*). The models of the NrdI system comprised the FMN isoalloxazine and ribityl moieties, sidechains of residues Trp74, Thr43, Thr42, Ser69, and backbone of Thr42, Thr43, Gly44, Phe45, Gly46, Ser69, Gly70, Asn71, Met77, Phe78, Gly79, and by including five water molecules resolved in the crystal structure, comprising 169–170 atoms (*Figure 4—figure supplement 2A*). When applicable, free N–H or C=O backbone groups were removed from the selection. Model systems of the R2b–NrdI complex included the same FMN and NrdI groups as in the NrdI-only systems, as well as NrdI Met8 sidechain, and sidechain of R2b residues Met16, Phe17, Gln20, Ile197, Val200, and five water molecules resolved in the structure, yielding a system with 245–246 atoms (*Figure 4—figure supplement 2B*). Protein sidechains were cut at the Cα-Cβ, and saturated with hydrogens. During the geometry optimizations, the Cβ positions, as well as Cα positions of the protein backbone, and terminal parts of the ribityl tail of FMN were fixed to their experimental positions. Additional restraints were also introduced on residues from the R2b subunit to account for the interaction of the binding site. Geometry optimizations were performed at the B3LYP-D3/def2-SVP/$\varepsilon$ = 4 level of theory, and single point energies were re-evaluated at the B3LYP-D3/def2-TZVP/$\varepsilon$ = 4 level (*Becke, 1993*; *Grimme et al., 2010*; *Klamt and Schüürmann, 1993*; *Lee et al., 1988*; *Schäfer et al., 1992*). The bending angles of the flavin were defined as the dihedral angle between the plane formed by the pyrimidine moiety and N5, N10 atoms, with the plane formed by the benzene moiety and N5, N10 atoms and measured with UCSF Chimera (*Pettersen et al., 2004*). Strain energies, computed at the B3LYP-D3/def2-TZVP level, were evaluated as the difference between the isolated flavin group optimized in its protein environment and the isolated flavin, reoptimized without the protein environment. The systems were modelled in oxidized state (FMN$_{ox}$), the neutral semiquinone state (FMN$_{sq}$), and anionic hydroquinone state (FMN$_{hq}$). All DFT calculations were performed with TURBOMOLE v. 7.5 (*Balasubramani et al., 2020*).

## Figures

Molecular figures were prepared with the PyMOL Molecular Graphics System, Version 2.4.2 Schrödinger, LLC. The surface representation of the channel in *Figure 5—figure supplement 3* was generated with the 'Cavities and Pockets' function of PyMOL after removing all waters from the model and with the surface cavity radius set to −2 and the surface cavity cut-off to −6. *Figure 3* was designed with ChemDraw (PerkinElmer Informatics). The bar graph in *Figure 5A* was generated with GraphPad Prism, Version 9.2.0 for iOS (GraphPad Software, San Diego, CA, USA). All figures except *Figure 3* were designed and assembled in Affinity Designer (Serif (Europe) Ltd, Nottingham, UK).

## Data availability

## Acknowledgements

We thank Robert Bolotovsky, Iris D Young, and Lee J O'Riordan for help with data processing and the staff at LCLS and SACLA. We thank Kristine Grãve for input and helpful discussions during data analysis and manuscript writing. AMO, PA, and ABu were supported by Diamond Light Source, the UK Science and Technology Facilities Council (STFC), a jointly funded strategic award from the Wellcome Trust and the Biotechnology and Biological Sciences Research Council (102593 to James Naismith). The DOT instrument used in this research was funded by Department of Energy (DOE), Office of Science, Office of Basic Energy Sciences (BES), Division of Chemical Sciences, Geosciences, and Biosciences (to JK, JY, and VKY). XFEL data were collected under proposal LU50 at LCLS, SLAC, Stanford, USA, and under proposal 2017B8085 at BL2 of SACLA, Japan. The Rayonix detector used at LCLS was supported by the NIH grant S10 OD023453. Use of the LCLS, SLAC National Accelerator Laboratory, is supported by the U.S. DOE, Office of Science, BES, under contract no. DE-AC02-76SF00515. Computational resources were provided by the Swedish National Infrastructure for Computing (SNIC 2021/1-40, SNIC 2022/1-29).

## Additional information

### Funding

| Funder | Grant reference number | Author |
| --- | --- | --- |
| Vetenskapsrådet | 2017-04018 | Martin Högbom |
| Vetenskapsrådet | 2021-03992 | Martin Högbom |
| European Research Council | HIGH-GEAR 724394 | Martin Högbom |
| Knut och Alice Wallenbergs Stiftelse | 2017.0275 | Martin Högbom |
| Knut och Alice Wallenbergs Stiftelse | 2019.0436 | Martin Högbom |
| Knut och Alice Wallenbergs Stiftelse | 2019.0251 | Ville RI Kaila |
| National Institutes of Health | GM133081 | Kyle D Sutherlin |
| National Institutes of Health | GM117126 | Nicholas K Sauter |
| National Institutes of Health | GM55302 | Vittal K Yachandra |
| National Institutes of Health | GM110501 | Junko Yano |
| National Institutes of Health | GM126289 | Jan Kern |
| Wellcome Trust | Investigator Award 210734/Z/18/Z | Allen M Orville |
| Royal Society | Wolfson Fellowship RSWF\R2\182017 | Allen M Orville |

The funders had no role in study design, data collection, and interpretation, or the decision to submit the work for publication. For the purpose of Open Access, the authors have applied a CC BY public copyright license to any Author Accepted Manuscript version arising from this submission.

### Author contributions

Juliane John, Conceptualization, Data curation, Formal analysis, Investigation, Visualization, Methodology, Writing – original draft, Writing – review and editing, Developed the study, Produced and purified proteins, Developed the crystallization protocol, Performed LCLS experiment, Refined, analysed and interpreted crystal structures, Performed TXRF measurements; Oskar Aurelius, Conceptualization, Investigation, Methodology, Writing – review and editing, Developed the study, Performed LCLS experiment; Vivek Srinivas, Conceptualization, Investigation, Methodology, Developed the study, Performed SACLA experiment; Patricia Saura, Formal analysis, Investigation, Writing – review and editing, Performed and analysed the DFT calculations; In-Sik Kim, Investigation, Methodology, Developed, tested and ran the sample delivery system, Performed LCLS experiment; Asmit Bhowmick, Formal analysis, Investigation, Writing – review and editing, Processed and analysed XFEL data, Performed LCLS experiment; Philipp S Simon, Investigation, Methodology, Writing – review and editing, Developed, tested and ran the sample delivery system, Performed LCLS experiment; Medhanjali Dasgupta, Formal analysis, Writing – review and editing, Processed and analysed XFEL data; Cindy Pham, Investigation, Methodology, Developed, tested and ran the sample delivery system, Performed LCLS experiment; Sheraz Gul, Investigation, Methodology, Developed, tested and ran the sample delivery system, Performed LCLS experiment; Kyle D Sutherlin, Formal analysis, Funding acquisition, Processed and analysed XFEL data, Performed LCLS experiment; Pierre Aller, Investigation, Writing – review and editing, Performed LCLS and SACLA experiments; Agata Butryn, Formal analysis, Investigation, Writing – review and editing, Processed and analysed XFEL data, Performed LCLS experiment;

Allen M Orville, Funding acquisition, Investigation, Methodology, Writing – review and editing, Developed, tested and ran the sample delivery system, Performed LCLS and SACLA experiments; Mun Hon Cheah, Investigation, Writing – review and editing, Performed SACLA experiment; Shigeki Owada, Investigation, Performed SACLA experiment; Kensuke Tono, Investigation, Performed SACLA experiment; Franklin D Fuller, Investigation, Methodology, Developed, tested and ran the sample delivery system, Operated the MFX instrument at LCLS,Performed LCLS and SACLA experiments; Alexander Batyuk, Investigation, Operated the MFX instrument at LCLS, Performed LCLS experiment; Aaron S Brewster, Formal analysis, Investigation, Processed and analysed XFEL data, Performed LCLS experiment; Nicholas K Sauter, Formal analysis, Funding acquisition, Writing – review and editing, Processed and analysed XFEL data, Performed LCLS experiment; Vittal K Yachandra, Resources, Funding acquisition, Investigation, Performed LCLS and SACLA experiments; Junko Yano, Resources, Funding acquisition, Investigation, Writing – review and editing, Performed LCLS and SACLA experiments; Ville RI Kaila, Formal analysis, Funding acquisition, Investigation, Writing – review and editing, Performed and analysed the DFT calculations; Jan Kern, Resources, Funding acquisition, Investigation, Methodology, Project administration, Writing – review and editing, Developed, tested and ran the sample delivery system, Performed LCLS and SACLA experiments; Hugo Lebrette, Conceptualization, Formal analysis, Supervision, Investigation, Methodology, Writing – original draft, Project administration, Writing – review and editing, Developed the study, Produced and purified proteins, Developed the crystallization protocol, Performed SACLA and LCLS experiments, Interpreted crystal structures; Martin Högbom, Conceptualization, Resources, Supervision, Funding acquisition, Investigation, Methodology, Project administration, Writing – review and editing, Developed the study, Performed LCLS and SACLA experiments, Interpreted crystal structures

### Author ORCIDs
Juliane John http://orcid.org/0000-0001-9626-3670
Vivek Srinivas http://orcid.org/0000-0002-0265-1873
Patricia Saura http://orcid.org/0000-0003-2575-9913
Philipp S Simon http://orcid.org/0000-0002-2859-4475
Agata Butryn http://orcid.org/0000-0002-5227-4770
Allen M Orville http://orcid.org/0000-0002-7803-1777
Mun Hon Cheah http://orcid.org/0000-0001-5732-1524
Shigeki Owada http://orcid.org/0000-0002-1451-7612
Kensuke Tono http://orcid.org/0000-0003-1218-3759
Alexander Batyuk http://orcid.org/0000-0002-9393-2880
Aaron S Brewster http://orcid.org/0000-0002-0908-7822
Nicholas K Sauter http://orcid.org/0000-0003-2786-6552
Junko Yano http://orcid.org/0000-0001-6308-9071
Ville RI Kaila http://orcid.org/0000-0003-4464-6324
Jan Kern http://orcid.org/0000-0002-7272-1603
Hugo Lebrette http://orcid.org/0000-0002-8081-181X
Martin Högbom http://orcid.org/0000-0001-5574-9383

### Decision letter and Author response
Decision letter https://doi.org/10.7554/eLife.79226.sa1
Author response https://doi.org/10.7554/eLife.79226.sa2

## Additional files

### Supplementary files
• Supplementary file 1. List of contacts that prevent flavin mononucleotide (FMN) from bending due to R2b–NrdI complex formation. The contacts of FMN to NrdI are not listed since those are also formed in the absence of R2b. The only additional contacts formed with between FMN and R2b are with the side chains (sc) of Phe17 and Gln20. The contacts formed by those side chains are also given. Contacts in both the oxidized and reduced structures are listed.

• Supplementary file 2. Bending angle (in degrees) and strain energies ($E_{strain}$, in eV) of the isoalloxazine ring of flavin mononucleotide (FMN) in *Bc*NrdI based on density functional theory (DFT) calculations. The strain energies are computed relative to the same redox state of the flavin, optimized without the protein surroundings. See Materials and methods for definition of the

bending angle.

• Supplementary file 3. Redox potential shifts of NrdI and R2b–NrdI. The relative redox potentials, $\Delta E_{tot}$, are computed at the B3LYP-D3/def2-TZVP/$\varepsilon$ = 4 level relative to the same redox transition of the isolated flavin in $\varepsilon$ = 4. The shifts in strain energy (at $\varepsilon$ = 4), $\Delta E_{strain}$, refer to the same redox transition relative to the isolated flavin, optimized without surroundings. The protein electrostatic shift, $\Delta E_{el}$, are estimated based on the difference between $\Delta E_{tot}$ and $\Delta E_{strain}$ ($\Delta E_{tot} = \Delta E_{strain} + \Delta E_{el}$). $\Delta\Delta E_{tot}$ are the total shifts in redox potentials of R2b–NrdI relative to NrdI alone.

• MDAR checklist

### Data availability

The atomic coordinates and crystallographic data have been deposited in the Protein Data Bank under the following accession codes: 7Z3D and 7Z3E.

The following datasets were generated:

| Author(s) | Year | Dataset title | Dataset URL | Database and Identifier |
|---|---|---|---|---|
| John J, Lebrette H, Aurelius O, Högbom M | 2022 | XFEL structure of Class Ib ribonucleotide reductase dimanganese(II) NrdF in complex with oxidized NrdI from Bacillus cereus | https://www.rcsb.org/structure/7Z3D | RCSB Protein Data Bank, 7Z3D |
| John J, Lebrette H, Aurelius O, Högbom M | 2022 | XFEL structure of Class Ib ribonucleotide reductase dimanganese(II) NrdF in complex with hydroquinone NrdI from Bacillus cereus | https://www.rcsb.org/structure/7Z3E | RCSB Protein Data Bank, 7Z3E |

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
