## [Editor Report]

This paper reports a fundamental set of new results that are obtained using compelling methods in protein crystallography and related fields to investigate and visualize the complex mechanism of an enzyme. The paper will be of interest to a broad audience in structural biology, biochemistry, and enzymology, providing a detailed mechanism of an important biological system and demonstrating useful tools. The work is timely and has implications for future investigations of complex biochemical processes.

---

## [Decision Letter]

**Decision letter after peer review:**

Thank you for submitting your article "Redox-controlled structural reorganization and flavin strain within the ribonucleotide reductase R2b-NrdI complex monitored by serial femtosecond crystallography" for consideration by *eLife*. Your article has been reviewed by 3 peer reviewers, and the evaluation has been overseen by a Reviewing Editor and Volker Dötsch as the Senior Editor. The reviewers have opted to remain anonymous.

Essential revisions:

1) Dark sq-bound crystals were shown in Figure S1, but no preparation methods and results were mentioned regarding these crystals. Was the sq radical stable in the crystalline form? And did these crystals yield valuable information? The sq-bound structure could bridge the transition from the fully reduced to the oxidized states.

2) The manuscript mentioned multiple FMN conformations of various oxidation and complexation states, while Figure 2A missed one critical conformation, the theoretical calculation of FMNsq (-3.9{degree sign} bending angle). It is suggested to include this conformation or replacing the sq- (physiologically irrelevant form) in Figure 2A.

3) When comparing the flavin conformation with other published data and describing the structural reorganization at the interface, two highly related references reporting the complex structures from *E. coli* and *B. cereus* (Boal et al., 2010; Hammerstad et al., 2014) were not included in the comparison and discussion.

4) In this study, the oxidized crystals were reduced by dithionite to yield the hydroquinone-bound structure. The fixed crystal lattice may restrict the structural movement upon the chemical reduction. Have the authors tried to first generate the reduced form in the solution state and then crystallize the complex?

5) If the redox state of FMN indeed controls the structural reorganization, synchrotron structures of the crystals prepared with oxidized flavin obtained should correspond to a reduced state, consistent with the R2b-NrdIhq structure. Showing such a structure would be a necessary control experiment to support the redox-controlled reorganization as claimed and exclude the possibility that the observed structural movement is due to the condition variation introduced by adding 1 M dithionite to the oxidized crystals.

6) The authors suggested that the unbending conformation reduces the redox potential of hydroquinone, but no direct evidence supports this statement. If it is known knowledge, please supply citations. Otherwise, a computational study predicting the redox potentials of FMN of different bending angles would be helpful.

7) The manuscript could be strengthened by the inclusion of experiments to measure the redox potential of the flavin before and after R2b-Nrd1 binding to provide experimental evidence towards the claim that this binding event lowers the redox potential (or, alternatively, the authors could dial back their interpretation here (p. 24)). Since the different redox states all have distinct optical properties, it seems like this could be monitored relatively straightforwardly spectroscopically.

8) The authors did not take a firm position on the nature of the strong electron density feature near to FMN that appears to occupy a different position in the oxidized versus reduced structures. Whatever this is, it is positioned appropriately to play a key role in the mechanism of this system. It is very much appreciated that the authors took the time to model these electron density features with candidate atoms (i.e. water, chloride ion, superoxide radical) and presented this in the Supplementary Information and that they wrote that they could not be sure of the nature of this unknown entity. However, with their acquired level of molecular detail and knowledge of the mechanism of this enzyme complex, it is suggested that the authors take a stronger position on the nature of this entity. They could present this as a model of the potential role of this entity in the mechanism. Could this not represent a superoxide ion that was formed before reduction by sodium dithionite?

9) An important aspect of the work is observation of conformational changes (or lack thereof) in residues from both the R2b and Nrdl subunits as a result of the different redox states of FMN. On this front, a request is for the authors to do a deeper analysis into the residues of R2b subunit that hold the FMN ring in place, preventing it from bending (i.e. Met16, Phe17, Ile197, Val200, Gln20). What prevents these residues from changing conformation should FMN bend? Are these residues themselves held in place through a network of interactions? i.e. is there a second shell of important interactions involving these residues? On the Nrdl side, it is recommended that they present 2Fo-Fc electron density maps of its residues that change conformation between the redox states of FMN; this would give more confidence in accepting their statements about rotations and conformational changes. i.e. residues 43 to 46.

10) It is noticed that in PDB files 7Z3D and 7Z3E corresponding to the oxidized state, an unknown density was modeled with water molecules (HOH556 or HOH323, respectively)). This is clearly wrong, as there is significant positive Fo-Fc density for these in both structures (7.09 and 5.47 rmsd in Coot, respectively) here. Depending on what the authors decide to do regarding modeling this or not, if they decide to leave it as "unknown", the proper way to model this in the structure is with an "UNX" atom. https://www.rcsb.org/ligand/UNX

11) An important aspect of the mechanism of this complex is the tunnel through which the superoxide would be transferred from FMN to the di-divalent cation center of R2b. A surface representation of the tunnel would be a nice way to show it, but it was only shown in cartoon and stick representation in Supp. 3, or the schematic Schemes 1 and 3. It would be nice to see a second panel added to Supp. 3 to show a surface representation of tunnel (It is recommended to use the Cavities and Pockets function in Pymol).

12) The high resolution cutoff of the x-ray diffraction data collection was too generous. For both structures at 2.0 A resolution as the high resolution, mean I/σ(I) values were below 1 and the CC1/2 values were below 0.5, which are the typical cutoffs for a highest resolution shell in the literature. Similarly, the Rsplit values were quite high at these highest resolution shell. It is suggested that the authors justify their choice of 2.0 A or adjust to meet the typically accepted cutoffs.

13) The authors claim that both manganese ions are in the II state and refer the reader to Materials and methods and Discussion for details, but then do not indicate if their reduction protocol would reduce Mn(III) to Mn(II) or elaborate in the Discussion section. The authors are encouraged to address this.

---

## [Author Response]

Essential revisions:1) Dark sq-bound crystals were shown in Figure S1, but no preparation methods and results were mentioned regarding these crystals. Was the sq radical stable in the crystalline form? And did these crystals yield valuable information? The sq-bound structure could bridge the transition from the fully reduced to the oxidized states.

We agree with the reviewers that a sq-bound structure could yield valuable information, unfortunately, we do not have the corresponding dataset. Crystals shown in this figure (now Figure 2 – Figure suppl. 1) were obtained during optimization of the reduction protocol. Due to the instability of the sq state, it has not been possible to obtain a sufficiently homogeneous sample where we can accurately assess the exact oxidation state of the FMN. We have added an explanation for the production of the dark sq crystals in the legend of Figure 2 – Figure suppl. 1.

2) The manuscript mentioned multiple FMN conformations of various oxidation and complexation states, while Figure 2A missed one critical conformation, the theoretical calculation of FMNsq (-3.9{degree sign} bending angle). It is suggested to include this conformation or replacing the sq- (physiologically irrelevant form) in Figure 2A.

We thank the reviewers for this suggestion. We have modified this figure (now Figure 4) accordingly and added the missing angle of FMNsq. We have also performed density functional theory (DFT) calculations and modified the figure in order to include our new data. Moreover, we have remeasured the angles of FMN in our SFX structures using, for consistency, the same software as for measurement of angles in the DFT calculations (i.e., Chimera). Accordingly, the manuscript has been updated with the new values, which are 0.2° and 0.1° for *Bc*R2b_MnMn_-NrdI_ox_ and *Bc*R2b_MnMn_-NrdI_hq_, respectively (instead of -1° and 1.5° previously).

3) When comparing the flavin conformation with other published data and describing the structural reorganization at the interface, two highly related references reporting the complex structures from *E. coli* and *B. cereus* (Boal et al., 2010; Hammerstad et al., 2014) were not included in the comparison and discussion.

We did not include the complex structures from *E. coli* and *B. cereus* in the comparison and discussion on the FMN site as we believe these structures are not suitable for direct comparison. Firstly, these structures were obtained at synchrotron sources and therefore the extent of photoreduction and the oxidation state of FMN are unknown. This aspect is not discussed in the corresponding publications. In the publications reporting structures of NrdI from *B. cereus* (Røhr et al., 2010) and *B. anthracis*, (Johansson et al., 2010) on the other hand, this aspect is discussed and both are included for comparison in our manuscript. Secondly, the FMN cofactor present in the complex structures is always refined in the oxidized electronic state (PDB ligand ID: FMN) even when referred to as reduced. It would appear that the reduced form of the ligand would be a more suitable model to use during refinement (PDB ligand ID: FNR). As no mention of a refinement protocol specific for reduced FMN could be found in the corresponding publications the rationale for performing the refinements in this way is unclear and it appears likely that the geometric restraints used for refinement of the FMN cofactor are inappropriate, making comparisons problematic.

4) In this study, the oxidized crystals were reduced by dithionite to yield the hydroquinone-bound structure. The fixed crystal lattice may restrict the structural movement upon the chemical reduction. Have the authors tried to first generate the reduced form in the solution state and then crystallize the complex?

We have tried the approach suggested by the reviewers. Unfortunately, we were not able to produce crystals of the complex in the reduced form as the dithionite concentration was either too low to prevent NrdI reoxidation or too high to not be detrimental to crystallization. However, there are no crystal contacts in the direct vicinity of the FMN binding site and thus we think it is unlikely that they could restrict any conformational changes of the cofactor and its immediate environment.

5) If the redox state of FMN indeed controls the structural reorganization, synchrotron structures of the crystals prepared with oxidized flavin obtained should correspond to a reduced state, consistent with the R2b-NrdIhq structure. Showing such a structure would be a necessary control experiment to support the redox-controlled reorganization as claimed and exclude the possibility that the observed structural movement is due to the condition variation introduced by adding 1 M dithionite to the oxidized crystals.

Exposing a crystal to a synchrotron beam indeed reduces the structure but the extent of the reduction is not known, since this is dependent on several factors (beam intensity, crystal properties, solvent content, ligand properties). Also, the structural changes accompanying redox changes at cryogenic temperatures may or may not represent what happens at room temperature. For this reason, we do not agree that the proposed experiment would be an appropriate control. While we understand the concern that the presence of dithionite might influence the results, the used concentration for reduction is 20 mM and it is also washed from the crystals before data collection (by washing the crystals 3 times with dithionite-free buffer). For these reasons we find the current setup to be a more appropriate control. We realized that the description of the protocol was not clear regarding the dithionite concentration – the crystals were at no point subjected to 1 M dithionite, this refers to the stock solution. We have modified the section “Reduction protocol” in the Materials and methods to clarify the protocol.

6) The authors suggested that the unbending conformation reduces the redox potential of hydroquinone, but no direct evidence supports this statement. If it is known knowledge, please supply citations. Otherwise, a computational study predicting the redox potentials of FMN of different bending angles would be helpful.

The principles behind redox tuning by conformational strain are well established. For a discussion specifically regarding flavins, see e.g. Walsh and Miller (2003) where they show that the flavin bending conformation enforced by the protein influences the stability of the cofactor, and that planarizing the hydroquinone flavin lowers its ionization potential (https://doi.org/10.1016/S0166-1280(02)00719-4). This reference is cited in our manuscript. Moreover, we have now performed density functional theory calculations confirming that the FMN is under strain in the R2b-NrdI complex and that redox potentials are shifted.

7) The manuscript could be strengthened by the inclusion of experiments to measure the redox potential of the flavin before and after R2b-Nrd1 binding to provide experimental evidence towards the claim that this binding event lowers the redox potential (or, alternatively, the authors could dial back their interpretation here (p. 24)). Since the different redox states all have distinct optical properties, it seems like this could be monitored relatively straightforwardly spectroscopically.

We agree with the reviewer that additional results would strengthen our conclusions. We have therefore performed density functional theory calculations to estimate the change in strain energy and additional effects on the FMN redox properties upon R2b binding. Accordingly, we have included the following in the manuscript: a new paragraph in the Results, a couple of sentences in the Discussion, a new section in the Materials and methods, an update of a main figure (now Figure 4), a new supplementary figure (Figure 4 – Figure suppl. 2), and new supplementary tables (Supplementary Files 2 and 3). Finally, we have removed “E_FMN_” from the last scheme (now Figure 7).

8) The authors did not take a firm position on the nature of the strong electron density feature near to FMN that appears to occupy a different position in the oxidized versus reduced structures. Whatever this is, it is positioned appropriately to play a key role in the mechanism of this system. It is very much appreciated that the authors took the time to model these electron density features with candidate atoms (i.e. water, chloride ion, superoxide radical) and presented this in the Supplementary Information and that they wrote that they could not be sure of the nature of this unknown entity. However, with their acquired level of molecular detail and knowledge of the mechanism of this enzyme complex, it is suggested that the authors take a stronger position on the nature of this entity. They could present this as a model of the potential role of this entity in the mechanism. Could this not represent a superoxide ion that was formed before reduction by sodium dithionite?

We appreciate the comment and agree that we have been rather careful in the interpretation. Given the experimental conditions we do not think superoxide could be present in the current structure while chloride is present at 100 mM in the crystallization condition. Thus, chloride seems the most likely candidate. Still, given the similar charge and size of the molecule the binding site may well be relevant for superoxide binding and gating, as the referee points out. We have updated the Discussion to more clearly emphasize this point.

9) An important aspect of the work is observation of conformational changes (or lack thereof) in residues from both the R2b and Nrdl subunits as a result of the different redox states of FMN. On this front, a request is for the authors to do a deeper analysis into the residues of R2b subunit that hold the FMN ring in place, preventing it from bending (i.e. Met16, Phe17, Ile197, Val200, Gln20). What prevents these residues from changing conformation should FMN bend? Are these residues themselves held in place through a network of interactions? i.e. is there a second shell of important interactions involving these residues? On the Nrdl side, it is recommended that they present 2Fo-Fc electron density maps of its residues that change conformation between the redox states of FMN; this would give more confidence in accepting their statements about rotations and conformational changes. i.e. residues 43 to 46.

We thank the reviewers for this important comment. The main interactions are between the outermost methyl groups of FMN and R2b residues Phe17 and Gln20, which are also restricted in their movement by surrounding residues. We have added a more detailed analysis regarding this issue by modifying the corresponding section in the Results and by including a list of interactions in a supplementary table (Supplementary File 1). Furthermore, we have added a supplementary figure (Figure 4 – Figure suppl. 3) showing the 2Fo-Fc and Fo-Fc electron density maps for NrdI residues 43 to 46.

10) It is noticed that in PDB files 7Z3D and 7Z3E corresponding to the oxidized state, an unknown density was modeled with water molecules (HOH556 or HOH323, respectively)). This is clearly wrong, as there is significant positive Fo-Fc density for these in both structures (7.09 and 5.47 rmsd in Coot, respectively) here. Depending on what the authors decide to do regarding modeling this or not, if they decide to leave it as "unknown", the proper way to model this in the structure is with an "UNX" atom. https://www.rcsb.org/ligand/UNX

We agree with the reviewers and because we prefer to leave this density as “unknown”, we have now replaced the previously modeled water molecule into an “UNX” atom in the two structures. Accordingly, we have mentioned this point in the Materials and methods.

11) An important aspect of the mechanism of this complex is the tunnel through which the superoxide would be transferred from FMN to the di-divalent cation center of R2b. A surface representation of the tunnel would be a nice way to show it, but it was only shown in cartoon and stick representation in Supp. 3, or the schematic Schemes 1 and 3. It would be nice to see a second panel added to Supp. 3 to show a surface representation of tunnel (It is recommended to use the Cavities and Pockets function in Pymol).

We agree that a surface representation is a nicer way to visualize the tunnel. As suggested, we have added a second panel to the supplementary figure (Figure 5 – Figure suppl. 2B) with the surface representation of the tunnel and we have updated the corresponding section of the Materials and methods to describe how the figure was prepared using the Cavities and Pockets function in PyMOL.

12) The high resolution cutoff of the x-ray diffraction data collection was too generous. For both structures at 2.0 A resolution as the high resolution, mean I/σ(I) values were below 1 and the CC1/2 values were below 0.5, which are the typical cutoffs for a highest resolution shell in the literature. Similarly, the Rsplit values were quite high at these highest resolution shell. It is suggested that the authors justify their choice of 2.0 A or adjust to meet the typically accepted cutoffs.

We thank the reviewers for this suggestion. For the two R2b-NrdI datasets presented in this paper, the CC_1/2_, *R*_split_ and mean I/σ(I) values are ~0.3, ~0.8 and ~0.7, respectively. While it is true that the mean I/σ(I) value is lower than 1.0 in our case, we argue that the I/σ(I) value relies largely on the error models chosen, as discussed in Brewster et al., (2019)^1^ and Karplus and Diedrichs (2015)^2^, and therefore should not be used to determine resolution cutoffs. The *R*_split_ value in general is an unweighted sum which will undermine final data quality if data of varying quality are merged into a dataset, as stated in Karplus and Diedrichs (2015)^2^. Moreover, *R*_split_ values depend dominantly on signal bandwidth^3^, and for these reasons we do not consider *R*_split_ to be a reliable parameter to decide resolution cutoff either.

Typically, we use two different criteria to decide the final resolution cutoff for a particular dataset:

a) The multiplicity cutoff where we cut the data where it falls off the 10-fold multiplicity value.

b) CC_1/2_ values where the value of these parameters do not monotonically decrease as a function of resolution anymore.

The datasets at 2.0 Å fulfill both the standards we typically use for our XFEL datasets (see e.g., Kern 2018, Ibrahim 2020, Srinivas 2020, Rabe 2021)^4-7^, and therefore we think it is justified that we use the same criteria for these R2b-NrdI datasets. We have added a sentence to the Materials and methods to justify our choice for resolution cutoff.

1. Brewster *et al.* (2019) https://doi.org/10.1107/S2059798319012877

2. Karplus and Diedrichs (2015) https://doi.org/10.1016/j.sbi.2015.07.003

3. White *et al.* (2013) https://doi.org/10.1107/S0907444913013620

4. Kern *et al.* (2018) https://doi.org/10.1038/s41586-018-0681-2

5. Ibrahim *et al.* (2020) https://doi.org/10.1073/pnas.2000529117

6. Srinivas *et al.* (2020) https://doi.org/10.1021/jacs.0c05613

7. Rabe *et al.* (2021) https://doi.org/10.1126/sciadv.abh0250

13) The authors claim that both manganese ions are in the II state and refer the reader to Materials and methods and Discussion for details, but then do not indicate if their reduction protocol would reduce Mn(III) to Mn(II) or elaborate in the Discussion section. The authors are encouraged to address this.

The metal site in the R2b protein is in the Mn(II)Mn(II) state. R2b was purified in the metal-depleted form and reconstituted with Mn(II)Cl_2_ before crystallization with oxidised NrdI. The metal site is thus not subjected to oxidation by reduced NrdI in presence of O_2_. As has also been shown previously, the Mn(II)/Mn(II) metal site does not react directly with molecular oxygen. Moreover, the coordination of the metal ions in the crystal structures is fully consistent with the reduced (Mn(II)/Mn(II)) metal site when compared with previously published structures of class Ib R2. We have clarified the corresponding sections (i.e., “Crystallization” and “Reduction protocol") in the Materials and methods.